# MODEL TRANSFERABILITY WITH RESPONSIVE DECISION SUBJECTS

## ABSTRACT

This paper studies model transferability when human decision subjects respond to a deployed machine learning model. In our setting, an agent or a user corresponds to a sample $(X, Y)$ drawn from a distribution $\mathcal{D}$ and will face a model $h$ and its classification result $h(X)$. Agents can modify $X$ to adapt to $h$, which will incur a distribution shift on $(X, Y)$. Therefore, when training $h$, the learner will need to consider the subsequently "induced" distribution when the output model is deployed. Our formulation is motivated by applications where the deployed machine learning models interact with human agents, and will ultimately face *responsive* and *interactive* data distributions. We formalize the discussions of the transferability of a model by studying how the model trained on the available source distribution (data) would translate to the performance on the induced domain. We provide both upper bounds for the performance gap due to the induced domain shift, as well as lower bounds for the trade-offs that a classifier has to suffer on either the source training distribution or the induced target distribution. We provide further instantiated analysis for two popular domain adaptation settings with *covariate shift* and *target shift*.

## 1 INTRODUCTION

Decision makers are increasingly required to be transparent on their decision making to offer the "right to explanation" (Goodman & Flaxman, 2017; Selbst & Powles, 2018; Ustun et al., 2019) [1]. Being transparent also invites potential adaptations from the population, leading to potential shifts. We are motivated by settings where the deployed machine learning models interact with human agents, which will ultimately face data distributions that reflect how human agents respond to the models. For instance, when a model is used to decide loan applications, candidates may adapt their features based on the model specification in order to maximize their chances of approval; thus the loan decision classifier observes a data distribution caused by its own deployment (e.g., see Figure 1 for a demonstration). Similar observations can be articulated for application in insurance sector (i.e. developing policy s.t. customers' behaviors might adapt to lower premium (Haghtalab et al., 2020)), education sector (i.e. developing courses when students are less incentivized to cheat (Kleinberg & Raghavan, 2020)) and so on.

| FEATURE | WEIGHT | ORIGINAL VALUE | | ADAPTED VALUE |
|---|---|---|---|---|
| Income | 2 | $ 6,000 | $\longrightarrow$ | $ 6,000 |
| Education Level | 3 | College | $\longrightarrow$ | College |
| Debt | **-10** | $40,000 | $\longrightarrow$ | **$20,000** |
| Savings | **5** | $20,000 | $\longrightarrow$ | **$0** |

Figure 1: An example of an agent who originally has both savings and debt, observes that the classifier penalizes debt (weight -10) more than it rewards savings (weight +5), and concludes that their most efficient adaptation is to use their savings to pay down their debt.

This paper investigates model transferability when the underlying distribution shift is induced by the deployed model. What we would like to have is some guarantee on the *transferability* of a classifier —

---

[1]See Appendix A.1 for more detailed discussions.

that is, how training on the available source distribution $\mathcal{D}_S$ translates to performance on the induced domain $\mathcal{D}(h)$, which depends on the model $h$ being deployed. A key concept in our setting is the *induced risk*, defined as the error a model incurs on the distribution induced by itself:

$$\text{Induced Risk}: \quad \text{Err}_{\mathcal{D}(h)}(h) := \mathbb{P}_{\mathcal{D}(h)}(h(X) \neq Y) \tag{1}$$

Most relevant to the above formulation is the strategic classification literature (Hardt et al., 2016a; Chen et al., 2020b). In this literature, agents are modeled as rational utility maximizers and game theoretical solutions were proposed to characterize the induced risk. However, our results are motivated by the following challenges in more general scenarios:

- **Modeling assumptions being restrictive** In many practical situations, it is often hard to faithfully characterize agents' utilities. Furthermore, agents might not be fully rational when they response. All the uncertainties can lead to a far more complicated distribution change in $(X, Y)$, as compared to often-made assumptions that agents only change $X$ but not $Y$ (Chen et al., 2020b).

- **Lack of access to response data** Another relevant literature to our work is performative prediction (Perdomo et al., 2020). In performative prediction, one would often require knowing $\mathcal{D}(h)$ or having samples observed from $\mathcal{D}(h)$ through repeated experiments. We posit that machine learning practitioners may only have access to data from the source distribution during training, and although they anticipate changes in the population due to human agents' responses, they cannot observe this new distribution until the model is actually deployed.

- **Retraining being costly** Even when samples from the induced data distribution are available, retraining the model from scratch may be impractical due to computational constraints.

The above observations motivate us to understand the transferability of a model trained on the source data to the domain induced by the deployment of itself. We study several fundamental questions:

- **Source risk $\Rightarrow$ Induced risk** For a given model $h$, how different is $\text{Err}_{\mathcal{D}(h)}(h)$, the error on the distribution induced by $h$, from $\text{Err}_{\mathcal{D}_S}(h) := \mathbb{P}_{\mathcal{D}_S}(h(X) \neq Y)$, the error on the source?

- **Induced risk $\Rightarrow$ Minimum induced risk** How much higher is $\text{Err}_{\mathcal{D}(h)}(h)$, the error on the induced distribution, than $\min_{h'} \text{Err}_{\mathcal{D}(h')}(h')$, the minimum achievable induced error?

- **Induced risk of *source optimal* $\Rightarrow$ Minimum induced risk** Of particular interest, and as a special case of the above, how does $\text{Err}_{\mathcal{D}(h_S^*)}(h_S^*)$, the induced error of the optimal model trained on the source distribution $h_S^* := \arg\min_h \text{Err}_{\mathcal{D}_S}(h)$, compare to $h_T^* := \arg\min_h \text{Err}_{\mathcal{D}(h)}(h)$?

- **Lower bound for learning tradeoffs** What is the minimum error a model must incur on either the source distribution $\text{Err}_{\mathcal{D}_S}(h)$ or its induced distribution $\text{Err}_{\mathcal{D}(h)}(h)$?

For the first three questions, we prove upper bounds on the additional error incurred when a model trained on a source distribution is transferred over to its induced domain. We also provide lower bounds for the trade-offs a classifier has to suffer on either the source training distribution or the induced target distribution. We then show how to specialize our results to two popular domain adaptation settings: *covariate shift* (Shimodaira, 2000; Zadrozny, 2004; Sugiyama et al., 2007; 2008; Zhang et al., 2013b) and *target shift* (Lipton et al., 2018; Guo et al., 2020; Zhang et al., 2013b). All omitted proofs can be found in the Appendix.

## 1.1 RELATED WORKS

Most relevant to us are three topics: strategic classification (Hardt et al., 2016a; Chen et al., 2020b; Dekel et al., 2010; Dong et al., 2018; Chen et al., 2020a; Miller et al., 2020; Kleinberg & Raghavan, 2020), a recently proposed notion of *performative prediction* (Perdomo et al., 2020; Mendler-Dünner et al., 2020), and domain adaptation (Jiang, 2008; Ben-David et al., 2010; Sugiyama et al., 2008; Zhang et al., 2019; Kang et al., 2019; Zhang et al., 2020).

Hardt et al. (2016a) pioneered the formalization of strategic behavior in classification based on a sequential two-player game between agents and classifiers. Subsequently, Chen et al. (2020b) addressed the question of repeatedly learning linear classifiers against agents who are strategically trying to game the deployed classifiers. Most of the existing literature focuses on finding the optimal classifier by assuming fully rational agents (and by characterizing the equilibrium response). In contrast, we do not make these assumptions and primarily study the transferability when only having knowledge of source data.

Our result was inspired by the transferability results in domain adaptations (Ben-David et al., 2010; Crammer et al., 2008; David et al., 2010). Later works examined specific domain adaptation models, such as covariate shift (Shimodaira, 2000; Zadrozny, 2004; Gretton et al., 2009; Sugiyama et al., 2008; Zhang et al., 2013b;a) and target/label shift (Lipton et al., 2018; Azizzadenesheli et al., 2019). A commonly established solution is to perform reweighted training on the source data, and robust and efficient solutions have been developed to estimate the weights accurately (Sugiyama et al., 2008; Zhang et al., 2013b;a; Lipton et al., 2018; Guo et al., 2020).

Our work, at the first sight, looks similar to several other area of studies. For instance, the notion of observing an "induced distribution" resembles similarity to the adversarial machine learning literature (Lowd & Meek, 2005; Huang et al., 2011; Vorobeychik & Kantarcioglu, 2018). One of the major differences between us and adversarial machine learning is the true label $Y$ stays the same for the attacked feature while in our paper, both $X$ and $Y$ might change in the adapted distribution $\mathcal{D}(h)$. In Appendix A.2, we provide detailed comparisons with some areas in domain adaptations, including domain generalization, adversarial attack and test-time adaptation. In particular, similar to domain generalization, one of the biggest challenge for our setting is the lack of access to data from the target distribution during training.

## 2 FORMULATION

Suppose we are learning a parametric model $h \in \mathcal{H}$ for a binary classification problem. Its training data set $S := \{x_i, y_i\}_{i=1}^N$ is drawn from a *source* distribution $\mathcal{D}_S$, where $x_i \in \mathbb{R}^d$ and $y_i \in \{-1, +1\}$. However, $h$ will then be deployed in a setting where the samples come from a *test* or *target* distribution $\mathcal{D}_T$ that can differ substantially from $\mathcal{D}_S$. Therefore instead of minimizing the prediction error on the source distribution $\mathrm{Err}_{\mathcal{D}_S}(h) := \mathbb{P}_{\mathcal{D}_S}(h(X) \neq Y)$, the goal is to find $h^*$ that minimizes $\mathrm{Err}_{\mathcal{D}_T}(h) := \mathbb{P}_{\mathcal{D}_T}(h(X) \neq Y)$. This is often referred to as the *domain adaptation problem*, where typically, the transition from $\mathcal{D}_S$ to $\mathcal{D}_T$ is assumed to be independent of the model $h$ being deployed.

We consider a setting in which the distribution shift depends on $h$, or is thought of as being *induced* by $h$. We will use $\mathcal{D}(h)$ to denote the *induced domain* by $h$:

$$\mathcal{D}_S \quad \rightarrow \quad encounters\ model\ h \quad \rightarrow \quad \mathcal{D}(h)$$

Strictly speaking, the induced distribution is a function of both $\mathcal{D}_S$ and $h$ and should be better denoted by $\mathcal{D}_S(h)$. To ease the notation, we will stick with $\mathcal{D}(h)$, but we shall keep in mind of its dependency of $\mathcal{D}_S$. For now, we do not restrict the dependency of $\mathcal{D}(h)$ of $\mathcal{D}$ and $h$, but later in Section 4 and 5 we will further instantiate $\mathcal{D}(h)$ under specific domain adaptation settings.

The challenge in the above setting is that when training $h$, the learner needs to carry the thoughts that $\mathcal{D}(h)$ should be the distribution it will be evaluated on and that the training cares about. Formally, we define the *induced risk* of a classifier $h$ as the 0-1 error on the distribution $h$ induces:

$$\text{Induced risk :} \quad \mathrm{Err}_{\mathcal{D}(h)}(h) := \mathbb{P}_{\mathcal{D}(h)}(h(X) \neq Y) \tag{2}$$

Denote by $h_T^* := \arg\min_{h \in \mathcal{H}} \mathrm{Err}_{\mathcal{D}(h)}(h)$ the classifier with minimum induced risk. More generally, when the loss may not be the 0-1 loss, we define the *induced $\ell$-risk* as

$$\text{Induced } \ell\text{-risk :} \quad \mathrm{Err}_{\ell, \mathcal{D}(h)}(h) := \mathbb{E}_{z \sim \mathcal{D}(h)}[\ell(h; z)]$$

The induced risks will be the primary quantities that we are interested in minimizing. The following additional notation will also be helpful:

- Distributions of $Y$ on a distribution $\mathcal{D}$: $\mathcal{D}_Y := \mathbb{P}_{\mathcal{D}}(Y = y)^2$, and in particular $\mathcal{D}_Y(h) := \mathbb{P}_{\mathcal{D}(h)}(Y = y)$, $\mathcal{D}_{Y|S} := \mathbb{P}_{\mathcal{D}_S}(Y = y)$.

- Distribution of $h$ on a distribution $\mathcal{D}$: $\mathcal{D}_h := \mathbb{P}_{\mathcal{D}}(h(X) = y)$, and in particular $\mathcal{D}_h(h) := \mathbb{P}_{\mathcal{D}(h)}(h(X) = y)$, $\mathcal{D}_{h|S} := \mathbb{P}_{\mathcal{D}_S}(h(X) = y)$.

- Marginal distribution of $X$ for a distribution $\mathcal{D}$: $\mathcal{D}_X := \mathbb{P}_{\mathcal{D}}(X = x)$, and in particular $\mathcal{D}_X(h) := \mathbb{P}_{\mathcal{D}(h)}(X = x)$, $\mathcal{D}_{X|S} := \mathbb{P}_{\mathcal{D}_S}(X = x)^3$.

- Total variation distance (Ali & Silvey, 1966): $d_{\mathrm{TV}}(\mathcal{D}, \mathcal{D}') := \sup_{\mathcal{O}} |\mathbb{P}_{\mathcal{D}}(\mathcal{O}) - \mathbb{P}_{\mathcal{D}'}(\mathcal{O})|$.

---

[2] The ":=" defines the RHS as the probability measure function for the LHS.

[3] For continuous $X$, the probability measure shall be read as the density function.

## 2.1 EXAMPLES OF DISTRIBUTION SHIFTS INDUCED BY MODEL DEPLOYMENT

We provide two example models to demonstrate the use cases for the distribution shift models described in our paper. We provide more details in Section 4.3 and Section 5.3.

**Strategic Classification** An example of distribution shift is when decision subjects perform *strategic response* to a decision rule. It is well-known that when human agents are subject to a decision rule, they will adapt their features so as to get a favorable prediction outcome. In the literature of strategic classification, we say the human agents perform strategic adaptation (Hardt et al., 2016a).

It is natural to assume that the feature distribution before and after the human agents' best response satisfies *covariate shift*: namely the feature distribution $\mathbb{P}(X)$ will change, but $\mathbb{P}(Y|X)$, the mapping between $Y$ and $X$, remain unchanged. Notice that this is different from the assumption made in the classic strategic classification setting Hardt et al. (2016a), which is to assume that the changes in the feature $X$ does not change the underlying true qualification $Y$. In our paper, we assume that changes in feature $X$ could potential lead to changes in the true qualification $Y$, and that the mapping between $Y$ and $X$ remains the same before and after the adaptation. This is a commonly assumption made in a recent line of work on incentivizing improvement behaviors from human agents(see, e.g. Chen et al. (2020a); Shavit et al. (2020)). We use Figure 2 (Left) as a demonstration of how distribution might shift for strategic response setting. In Section 4.3, we will use the strategic classification setup to verify our obtained results.

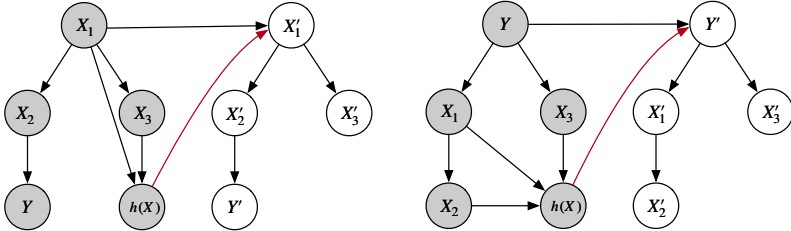

Figure 2: Example causal graph annotated to demonstrate covariate shift (**Left**) / target shift (**Right**) as a result of the deployment of $h$. Grey nodes indicate observable variables and transparent nodes are not observed at the training stage. Red arrow emphasises $h$ induces changes of certain variables.

**Replicator Dynamics** Replicator dynamics is a commonly used model to study the evolution of an adopted "strategy" in evolutionary game theory (Tuyls et al., 2006; Friedman & Sinervo, 2016; Taylor & Jonker, 1978; Raab & Liu, 2021). The core notion of it is the growth or decline of the population of each strategy depends on its "fitness". Consider the label $Y = \{-1, +1\}$ as the strategy, and the following behavioral response model to capture the induced target shift:

$$\frac{\mathbb{P}_{\mathcal{D}(h)}(Y = +1)}{\mathbb{P}_{\mathcal{D}_S}(Y = +1)} = \frac{\textbf{Fitness}(Y = +1)}{\mathbb{E}_{\mathcal{D}_S}[\textbf{Fitness}(Y)]}$$

In short, the change of the $Y = +1$ population depends on how predicting $Y = +1$ "fits" a certain utility function. For instance, the "fitness" can take the form of the prediction accuracy of $h$ for class +1: $\textbf{Fitness}(Y = +1) := \mathbb{P}_{\mathcal{D}_S}(h(X) = +1|Y = +1)$. Intuitively speaking, a higher "fitness" describes more success of agents who adopted a certain strategy ($Y = -1$ or $Y = +1$). Therefore, agents will imitate or replicate these successful peer agents by adopting the same strategy, resulting in an increase of the population ($\mathbb{P}_{\mathcal{D}(h)}(Y)$). With assuming $\mathbb{P}(X|Y)$ stays unchanged, this instantiates one example of a specific induced *target shift*. We will specify the condition for target shift in Section 5. We use Figure 2 (Right) as a demonstrating of how distribution might shift for the replicator dynamic setting. In Section 5.3, we will use a detailed replicator dynamics model to further instantiate our results.

## 3 TRANSFERABILITY OF LEARNING TO INDUCED DOMAINS

In this section, we first provide upper bounds for the transfer error of a classifier $h$ (that is, the difference between $\text{Err}_{\mathcal{D}(h)}(h)$ and $\text{Err}_{\mathcal{D}_S}(h)$), as well as between $\text{Err}_{\mathcal{D}(h)}(h)$ and $\text{Err}_{\mathcal{D}(h_T^*)}(h_T^*)$. We then provide lower bounds for $\max\{\text{Err}_{\mathcal{D}_S}(h), \text{Err}_{\mathcal{D}(h)}(h)\}$, that is, the minimum error a model $h$ must incur on either the source distribution $\mathcal{D}_S$ or the induced distribution $\mathcal{D}(h)$.

### 3.1 UPPER BOUND

We first investigate upper bounds for the transfer errors. We begin by showing generic bounds, and further instantiate the bound for specific domain adaptation settings in Section 4 and 5 . We begin with answering a central question in domain adaptation:

*How does a model $h$ trained on its training data set fare on the induced distribution $\mathcal{D}(h)$?*

To that end, define the minimum and $h$-dependent combined error of two distributions $\mathcal{D}$ and $\mathcal{D}'$ as:

$$\lambda_{\mathcal{D}\to\mathcal{D}'} := \min_{h'\in\mathcal{H}} \mathrm{Err}_{\mathcal{D}'}(h') + \mathrm{Err}_{\mathcal{D}}(h'), \ \Lambda_{\mathcal{D}\to\mathcal{D}'}(h) := \mathrm{Err}_{\mathcal{D}'}(h) + \mathrm{Err}_{\mathcal{D}}(h)$$

and $\mathcal{H}$-divergence as $d_{\mathcal{H}\times\mathcal{H}}(\mathcal{D},\mathcal{D}') = 2\sup_{h,h'\in\mathcal{H}}|\mathbb{P}_{\mathcal{D}}(h(X)\neq h'(X)) - \mathbb{P}_{\mathcal{D}'}(h(X)\neq h'(X))|$. The $\mathcal{H}$-divergence is celebrated measure proposed in the domain adaptation literature (Ben-David et al., 2010) which will be useful for bounding the difference in errors of two classifiers. Repeating classical arguments from Ben-David et al. (2010), we can easily prove the following:

**Theorem 3.1 (Source risk $\Rightarrow$ Induced risk).** *The difference between $Err_{\mathcal{D}(h)}(h)$ and $Err_{\mathcal{D}_S}(h)$ is upper bounded by: $Err_{\mathcal{D}(h)}(h) \leq Err_{\mathcal{D}_S}(h) + \lambda_{\mathcal{D}_S\to\mathcal{D}(h)} + \frac{1}{2}d_{\mathcal{H}\times\mathcal{H}}(\mathcal{D}_S,\mathcal{D}(h))$.*

The transferability of a model $h$ between $\mathrm{Err}_{\mathcal{D}(h)}(h)$ and $\mathrm{Err}_{\mathcal{D}_S}(h)$ looks precisely the same as in the classical domain adaptation setting (Ben-David et al., 2010). Nonetheless, an arguably more interesting quantity in our setting to understand is the difference between the induced error of a given model $h$ and the error induced by the optimal model $h_T^*$: $\mathrm{Err}_{\mathcal{D}(h)}(h) - \mathrm{Err}_{\mathcal{D}(h_T^*)}(h_T^*)$. We get the following bound, which differs from the one in Theorem 3.1:

**Theorem 3.2 (Induced risk $\Rightarrow$ Minimum induced risk).** *The difference between $Err_{\mathcal{D}(h)}(h)$ and $Err_{\mathcal{D}(h_T^*)}(h_T^*)$ is upper bounded by: $Err_{\mathcal{D}(h)}(h) - Err_{\mathcal{D}(h_T^*)}(h_T^*) \leq \frac{\lambda_{\mathcal{D}(h)\to\mathcal{D}(h_T^*)}+\Lambda_{\mathcal{D}(h)\to\mathcal{D}(h_T^*)}(h)}{2} + \frac{1}{2}\cdot d_{\mathcal{H}\times\mathcal{H}}(\mathcal{D}(h_T^*),\mathcal{D}(h))$.*

The above theorem informs us that the induced transfer error is bounded by the "average" achievable error on both distributions $\mathcal{D}(h)$ and $\mathcal{D}(h_T^*)$, as well as the $\mathcal{H}\times\mathcal{H}$ divergence between the two distributions. Reflecting on the difference between the bounds of Theorem 3.1 and Theorem 3.2, we see that the primary change is replacing the minimum achievable error $\lambda$ with the average of $\lambda$ and $\Lambda$.

### 3.2 LOWER BOUND

Now we provide a lower bound on the induced transfer error. We particularly want to show that at least one of the two errors $\mathrm{Err}_{\mathcal{D}_S}(h)$, and $\mathrm{Err}_{\mathcal{D}(h)}(h)$, must be lower-bounded by a certain quantity.

**Theorem 3.3 (Lower bound for learning tradeoffs ).** *Any model $h$ must incur the following error on either the source or induced distribution:* $\max\{Err_{\mathcal{D}_S}(h), Err_{\mathcal{D}(h)}(h)\} \geq \frac{d_{TV}(\mathcal{D}_{Y|S},\mathcal{D}_Y(h))-d_{TV}(\mathcal{D}_{h|S},\mathcal{D}_h(h))}{2}$.

The proof leverages the triangle inequality of $d_{\mathrm{TV}}$. This bound is dependent on $h$; however, by the data processing inequality of $d_{\mathrm{TV}}$ (and $f$-divergence functions in general) (Liese & Vajda, 2006), we have $d_{\mathrm{TV}}(\mathcal{D}_{h|S},\mathcal{D}_h(h)) \leq d_{\mathrm{TV}}(\mathcal{D}_{X|S},\mathcal{D}_X(h))$. Applying this to Theorem 3.3 yields:

**Corollary 3.4.** *For any model $h$,* $\max\{Err_{\mathcal{D}_S}(h), Err_{\mathcal{D}(h)}(h)\} \geq \frac{d_{TV}(\mathcal{D}_{Y|S},\mathcal{D}_Y(h))-d_{TV}(\mathcal{D}_{X|S},\mathcal{D}_X(h))}{2}$.

### 3.3 HOW TO USE OUR BOUNDS

The upper and lower bounds we derived in the previous sections (Theorem 3.2 and Theorem 3.3) depend on the following two quantities either explicitly or implicitly: 1) the distribution $\mathcal{D}(h)$ induced by the deployment of the model $h$ in question, and 2) the optimal target classifier $h_T^*$ as well as the distribution $\mathcal{D}(h_T^*)$ it induces. The bounds may therefore seem to be of only theoretical interest, since in reality we generally cannot compute $\mathcal{D}(h)$ without actual deployment, let alone compute $h_T^*$. Thus in general it is unclear how to compute the value of these bounds. Nevertheless, our bounds can still be useful and informative in the following ways:

**General modeling framework with flexible hypothetical shifting models** The bounds can be evaluated if the decision maker has a particular shift model in mind, which specifies how the population would adapt to a model. A common special case is when the decision maker posits an

individual-level agent response model (e.g. the strategic agent (Hardt et al., 2016a) - we demonstrate how to evaluate in Section 4.3). In these cases, the $H$-divergence can be consistently estimated from finite samples of the population (Wang et al., 2005), allowing the decision maker to estimate the performance gap of a given $h$ without deploying it. The general bounds provided can thus be viewed as a framework by which specialized, computationally tractable bounds can be derived.

**Estimate the optimal target classifier $h_T^*$ from a set of imperfect models** Secondly, when the decision maker has access to a set of imperfect models $\tilde{h}_1, \tilde{h}_2 \cdots \tilde{h}_t \in H^T$ that will predict a range of possible shifted distribution $\mathcal{D}(\tilde{h}_1), \cdots \mathcal{D}(\tilde{h}_t) \in \mathcal{D}^T$ and a range of possibly optimal target distribution $h_T \in \mathcal{H}^T$, the bounds on $h_T^*$ can be further instantiated by calculating the worst case in this predicted set [4]:

$$\text{Err}_{\mathcal{D}(h)}(h) - \text{Err}_{\mathcal{D}(h_T^*)}(h_T^*) \lesssim \max_{\mathcal{D}' \in \mathcal{D}^T, h' \in \mathcal{H}^T} \text{UpperBound}(\mathcal{D}', h'),$$

$$\max\{\text{Err}_{\mathcal{D}_S}(h), \text{Err}_{\mathcal{D}(h_T^*)}(h_T^*)\} \gtrsim \min_{\mathcal{D}' \in \mathcal{D}^T, h' \in \mathcal{H}^T} \text{LowerBound}(\mathcal{D}', h').$$

In addition, the challenge we are facing in this paper also shed lights on the danger of directly applying existing standard domain adaptation techniques when the shifting is caused by the deployment of the classifier itself, since the bound will depend on the resulting distribution as well. We add discussions on the tightness of our theoretical bounds in Appendix G.

## 4 COVARIATE SHIFT

In this section, we focus on a particular domain adaptation setting known as *covariate shift*, in which the distribution of features changes, but the distribution of labels conditioned on features does not:

$$\mathbb{P}_{\mathcal{D}(h)}(Y = y | X = x) = \mathbb{P}_{\mathcal{D}_S}(Y = y | X = x), \ \ \mathbb{P}_{\mathcal{D}(h)}(X = x) \neq \mathbb{P}_{\mathcal{D}_S}(X = x) \tag{3}$$

Thus with covariate shift, we have

$$\mathbb{P}_{\mathcal{D}(h)}(X = x, Y = y) = \mathbb{P}_{\mathcal{D}(h)}(Y = y | X = x) \cdot \mathbb{P}_{\mathcal{D}(h)}(X = x) = \mathbb{P}_{\mathcal{D}_S}(Y = y | X = x) \cdot \mathbb{P}_{\mathcal{D}(h)}(X = x)$$

Let $\omega_x(h) := \frac{\mathbb{P}_{\mathcal{D}(h)}(X=x)}{\mathbb{P}_{\mathcal{D}_S}(X=x)}$ be the *importance weight* at $x$, which characterizes the amount of adaptation induced by $h$ at instance $x$. Then for any loss function $\ell$ we have

**Proposition 4.1** (Expected Loss on $\mathcal{D}(h)$). $\mathbb{E}_{\mathcal{D}(h)}[\ell(h; X, Y)] = \mathbb{E}_{\mathcal{D}_S}[\omega_x(h) \cdot \ell(h; x, y)]$.

The above derivation was not new and offered the basis for performing importance reweighting when learning under coviarate shift (Sugiyama et al., 2008). The particular form informs us that $\omega_x(h)$ controls the generation of $\mathcal{D}(h)$ and encodes its dependency of both $\mathcal{D}_S$ and $h$, and is critical for deriving our results below.

### 4.1 UPPER BOUND

We now derive an upper bound for transferability under covariate shift. We will focus particularly on the optimal model trained on the source data $\mathcal{D}_S$, which we denote as $h_S^* := \arg\min_{h \in \mathcal{H}} \text{Err}_S(h)$. Recall that the classifier with minimum induced risk is denoted as $h_T^* := \arg\min_{h \in \mathcal{H}} \text{Err}_{\mathcal{D}(h)}(h)$. We can upper bound the difference between $h_S^*$ and $h_T^*$ as follows:

**Theorem 4.2** (Suboptimality of $h_S^*$). *Let $X$ be distributed according to $\mathcal{D}_S$. We have:*

$$\text{Err}_{\mathcal{D}(h_S^*)}(h_S^*) - \text{Err}_{\mathcal{D}(h_T^*)}(h_T^*) \leq \sqrt{\text{Err}_{\mathcal{D}_S}(h_T^*)} \cdot \left( \sqrt{\text{Var}(\omega_X(h_S^*))} + \sqrt{\text{Var}(\omega_X(h_T^*))} \right).$$

This result can be interpreted as follows: $h_T^*$ incurs an irreducible amount of error on the source data set, represented by $\sqrt{\text{Err}_{\mathcal{D}_S}(h_T^*)}$. Moreover, the difference in error between $h_S^*$ and $h_T^*$ is at its maximum when the two classifiers induce adaptations in "opposite" directions; this is represented by the sum of the standard deviations of their importance weights, $\sqrt{\text{Var}(\omega_X(h_S^*))} + \sqrt{\text{Var}(\omega_X(h_T^*))}$.

---

[4]UpperBound and LowerBound are the RHS expressions in Theorem 3.3 and Theorem 3.2.

## 4.2 LOWER BOUND

Recall from Theorem 3.3, for the general setting, it is unclear whether the lower bound is strictly positive or not. In this section, we provide further understanding for when the lower bound $\frac{d_{\text{TV}}(\mathcal{D}_{Y|S}, \mathcal{D}_Y(h)) - d_{\text{TV}}(\mathcal{D}_{h|S}, \mathcal{D}_h(h))}{2}$ is indeed positive under covariate shift. Under several assumptions, our previously provided lower bound in Theorem 3.3 is strictly positive with covariate shift.

**Assumption 4.3.** $|\mathbb{E}_{X \in X_+(h), Y=+1}[1 - \omega_X(h)]| \geq |\mathbb{E}_{X \in X_-(h), Y=+1}[1 - \omega_X(h)]|$.

where $X_+(h) = \{x : \omega_x(h) \geq 1\}$ and $X_-(h) = \{x : \omega_x(h) < 1\}$.

This assumption states that increased $\omega_x(h)$ value points are more likely to have positive labels.

**Assumption 4.4.** $|\mathbb{E}_{X \in X_+(h), h(X)=+1}[1 - \omega_X(h)]| \geq |\mathbb{E}_{X \in X_-(h), h(X)=+1}[1 - \omega_X(h)]|$.

This assumption states that increased $\omega_x(h)$ value points are more likely to be classified as positive.

**Assumption 4.5.** $\text{Cov}\big(\mathbb{P}_{\mathcal{D}_S}(Y = +1 | X = x) - \mathbb{P}_{\mathcal{D}_S}(h(x) = +1 | X = x), \omega_x(h)\big) > 0$.

This assumption is stating that for a classifier $h$, within all $h(X) = +1$ or $h(X) = -1$, a higher $\mathbb{P}_{\mathcal{D}}(Y = +1 | X = x)$ associates with a higher $\omega_x(h)$.

**Theorem 4.6.** *Assuming 4.3 - 4.5, the following lower bound is strictly positive for covariate shift:*

$$\max\{Err_{\mathcal{D}_S}(h), Err_{\mathcal{D}(h)}(h)\} \geq \frac{d_{TV}(\mathcal{D}_{Y|S}, \mathcal{D}_Y(h)) - d_{TV}(\mathcal{D}_{h|S}, \mathcal{D}_h(h))}{2} > 0.$$

## 4.3 EXAMPLE USING STRATEGIC CLASSIFICATION

As introduced in Section 2.1, we consider a setting caused by *strategic response* in which agents are classified by and adapt to a binary threshold classifier. In particular, each agent is associated with a $d$ dimensional continuous feature $x \in \mathbb{R}^d$ and a binary true qualification $y(x) \in \{-1, +1\}$, where $y(x)$ is a function of the feature vector $x$. Consistent with the literature in strategic classification (Hardt et al., 2016a), a simple case where after seeing the threshold binary decision rule $h(x) = 2 \cdot \mathbb{1}[x \geq \tau_h] - 1$, the agents will *best response* to it by maximizing the following utility function: $u(x, x') = h(x') - h(x) - c(x, x')$, where $c(x, x')$ is the *cost function* for decision subjects to modify their feature from $x$ to $x'$. We assume all agents are rational utility maximizers: they will only *attempt* to change their features when the benefit of manipulation is greater than the cost (i.e. when $c(x, x') \leq 2$) and agent will not change their feature if they are already accepted (i.e. $h(x) = +1$). For a given threshold $\tau_h$ and manipulation budget $B$, the theoretical best response of an agent with original feature $x$ is: $\Delta(x) = \arg \max_{x'} u(x, x')$ $s.t.$ $c(x, x') \leq B$. To make the problem tractable and meaningful, we further specify the following setups:

*Setup* 1. (Initial Feature) Agents' initial features are uniformly distributed between $[0, 1] \in \mathbb{R}^1$.

*Setup* 2. (Agent's Cost Function) The cost of changing from $x$ to $x'$ is proportional to the distance between them: $c(x, x') = \|x - x'\|$.

Setup 2 implies that only agents whose features are in between $[\tau_h - B, \tau_h)$ will *attempt* to change their feature. We also assume that feature updates are *probabilistic*, such that agents with features closer to the decision boundary $\tau_h$ have a greater *chance* of updating their feature and each updated feature $x'$ is sampled from a uniform distribution depending on $\tau_h$, $B$, and $x$ (see Setup 3 & 4):

*Setup* 3. (Agent's Success Manipulation Probability) For agents who *attempt* to update their features, the probability of a successful feature update is $\mathbb{P}(X' \neq X) = 1 - \frac{|x - \tau_h|}{B}$.

*Setup* 4 (Adapted Feature's Distribution). An agent's updated feature $x'$, given original $x$, manipulation budget $B$, and classification boundary $\tau_h$, is sampled as $X' \sim \text{Unif}(\tau_h, \tau_h + |B - x|)$.

Setup 4 aims to capture the fact that even though agent targets to change their feature to the decision boundary $\tau_h$ (i.e. the least cost action to get a favorable prediction outcome), they might end up reaching to a feature that is beyond the decision boundary. With the above setups, we can specify the bound in Theorem 4.2 for the strategic response setting as follows:

**Proposition 4.7** (Strategic Response Setting). $Err_{\mathcal{D}(h_S^*)}(h_S^*) - Err_{\mathcal{D}(h_T^*)}(h_T^*) \leq \sqrt{\frac{2B}{3} Err_{\mathcal{D}_S}(h_T^*)}$.

We can see that the upper bound for strategic response depends on the manipulation budget $B$, and the error the ideal classifier made on the source distribution $\text{Err}_{D_S}(h_T^*)$. This aligns with our intuition

that the smaller manipulation budget is, the less agents will change their features, thus leading to a tighter upper bound on the difference between $\text{Err}_{h_S^*}(h_S^*)$ and $\text{Err}_{h_T^*}(h_T^*)$. This bound also allows us to bound this quantity even without the knowledge of the mapping between $\mathcal{D}(h)$ and $h$, since we can directly compute $\text{Err}_{\mathcal{D}_S}(h_T^*)$ from the source distribution and an estimated optimal classifier $h_T^*$.

## 5 TARGET SHIFT

We consider another popular domain adaptation setting known as *target shift*, in which the distribution of labels changes, but the distribution of features conditioned on the label remains the same:

$$\mathbb{P}_{\mathcal{D}(h)}(X = x|Y = y) = \mathbb{P}_{\mathcal{D}_S}(X = x|Y = y), \ \ \mathbb{P}_{\mathcal{D}(h)}(Y = y) \neq \mathbb{P}_{\mathcal{D}_S}(Y = y) \tag{4}$$

For binary classification, let $p(h) := \mathbb{P}_{\mathcal{D}(h)}(Y = +1)$, and $\mathbb{P}_{\mathcal{D}(h)}(Y = -1) = 1 - p(h)$. Again, $p(h)$ encodes the induced adaptation from $\mathcal{D}_S$ and $h$. Then we have for any proper loss function $\ell$:

$$\mathbb{E}_{\mathcal{D}(h)}[\ell(h; X, Y)] = p(h) \cdot \mathbb{E}_{\mathcal{D}(h)}[\ell(h; X, Y)|Y = +1] + (1 - p(h)) \cdot \mathbb{E}_{\mathcal{D}(h)}[\ell(h; X, Y)|Y = -1]$$
$$= p(h) \cdot \mathbb{E}_{\mathcal{D}_S}[\ell(h; X, Y)|Y = +1] + (1 - p(h)) \cdot \mathbb{E}_{\mathcal{D}_S}[\ell(h; X, Y)|Y = -1]$$

We will adopt the following shorthands: $\text{Err}_+(h) := \mathbb{E}_{\mathcal{D}_S}[\ell(h; X, Y)|Y = +1]$, $\text{Err}_-(h) := \mathbb{E}_{\mathcal{D}_S}[\ell(h; X, Y)|Y = -1]$. Note that $\text{Err}_+(h), \text{Err}_-(h)$ are both defined on the conditional source distribution, which is invariant under the target shift assumption.

### 5.1 UPPER BOUND

We again upper bound the transferability of $h_S^*$ under target shift. Denote by $\mathcal{D}_+$ the positive label distribution on $\mathcal{D}_S$ ($\mathbb{P}_{\mathcal{D}_S}(X = x|Y = +1)$) and $\mathcal{D}_-$ the negative label distribution on $\mathcal{D}_S$ ($\mathbb{P}_{\mathcal{D}_S}(X = x|Y = -1)$). Let $p := \mathbb{P}_{\mathcal{D}_S}(Y = +1)$.

**Theorem 5.1.** *For target shift, the difference between $Err_{\mathcal{D}(h_S^*)}(h_S^*)$ and $Err_{\mathcal{D}(h_T^*)}(h_T^*)$ bounds as:*

$$Err_{\mathcal{D}(h_S^*)}(h_S^*) - Err_{\mathcal{D}(h_T^*)}(h_T^*) \leq |p(h_S^*) - p(h_T^*)| + (1 + p) \cdot (d_{TV}(\mathcal{D}_+(h_S^*), \mathcal{D}_+(h_T^*)) + d_{TV}(\mathcal{D}_-(h_S^*), \mathcal{D}_-(h_T^*))).$$

The above upper bound consists of two components. The first quantity captures the difference between the two induced distributions $\mathcal{D}(h_S^*)$ and $\mathcal{D}(h_T^*)$. The second quantity characterizes the difference between the two classifiers $h_S^*, h_T^*$ on the source distribution.

### 5.2 LOWER BOUND

Now we discuss lower bounds. Denote by $\text{TPR}_S(h)$ and $\text{FPR}_S(h)$ the true positive and false positive rates of $h$ on the source distribution $\mathcal{D}_S$. We prove the following:

**Theorem 5.2.** *For target shift, any model $h$ must incur the following error on either $\mathcal{D}_S$ or $\mathcal{D}(h)$:*

$$\max\{Err_{\mathcal{D}_S}(h), Err_{\mathcal{D}(h)}(h)\} \geq \frac{|p - p(h)| \cdot (1 - |TPR_S(h) - FPR_S(h)|)}{2}.$$

The proof extends the bound of Theorem 3.3 by further explicating each of $d_{\text{TV}}(\mathcal{D}_{Y|S}, \mathcal{D}_Y(h))$, $d_{\text{TV}}(\mathcal{D}_{h|S}, \mathcal{D}_h(h))$ under the assumption of target shift. Since $|\text{TPR}_S(h) - \text{FPR}_S(h)| < 0$ unless we have a trivial classifier that has either $\text{TPR}_S(h) = 1, \text{FPR}_S(h) = 0$ or $\text{TPR}_S(h) = 0, \text{FPR}_S(h) = 1$, the lower bound is strictly positive. Taking a closer look, the lower bound is determined linearly by how much the label distribution shifts: $p - p(h)$. The difference is further determined by the performance of $h$ on the source distribution through $1 - |\text{TPR}_S(h) - \text{FPR}_S(h)|$. For instance, when $\text{TPR}_S(h) > \text{FPR}_S(h)$, the quality becomes $\text{FNR}_S(h) + \text{FPR}_S(h)$, that is the more error $h$ makes, the larger the lower bound will be.

### 5.3 EXAMPLE USING REPLICATOR DYNAMICS

Let us instantiate the discussion using a specific fitness function for the replicator dynamics model (Section 2.1), which is the prediction accuracy of $h$ for class $y$:

$$\textbf{Fitness}(Y = y) := \mathbb{P}_{\mathcal{D}_S}(h(X) = y|Y = y) \tag{5}$$

Then we have $\mathbb{E}[\textbf{Fitness}(Y)] = 1 - \text{Err}_{\mathcal{D}_S}(h)$, and $\frac{p(h)}{\mathbb{P}_{\mathcal{D}_S}(Y = +1)} = \frac{\text{Pr}_{\mathcal{D}_S}(h(X) = +1|Y = +1)}{1 - \text{Err}_{\mathcal{D}_S}(h)}$. Plugging the result back to our Theorem 5.1 we have

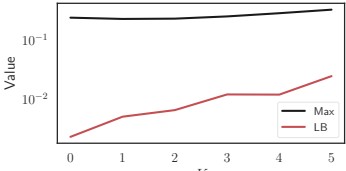 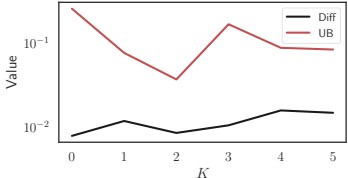

**Figure 3:** $\text{Diff} := \text{Err}_{\mathcal{D}(h_S^*)}(h_S^*) - \text{Err}_{\mathcal{D}(h_T^*)}(h_T^*)$, $\text{Max} := \max\{\text{Err}_{\mathcal{D}_S}(h_T^*), \text{Err}_{\mathcal{D}(h_T^*)}(h_T^*)\}$, $\text{UB} :=$ upper bound specified in Theorem 4.2, and $\text{LB} :=$ lower bound specified in Theorem 4.6. For each time step $K = k$, we compute and deploy the source optimal classifier $h_S^*$ and update the credit score for each individual according to the received decision as the new reality for time step $K = k + 1$. Details of the data generation is again deferred to Appendix C.

**Proposition 5.3.** *Under the replicator dynamics model in Eqn. (21), $|\omega(h_S^*) - \omega(h_T^*)|$ bounds as:*

$$|\omega(h_S^*) - \omega(h_T^*)| \leq \mathbb{P}_{\mathcal{D}_S}(Y = +1) \cdot \frac{|Err_{\mathcal{D}_S}(h_S^*) - Err_{\mathcal{D}_S}(h_T^*)| \cdot |TPR_S(h_S^*) - TPR_S(h_T^*)|}{(1 - Err_{\mathcal{D}_S}(h_S^*)) \cdot (1 - Err_{\mathcal{D}_S}(h_T^*))}.$$

That is, the difference between $\text{Err}_{\mathcal{D}(h_S^*)}(h_S^*)$ and $\text{Err}_{\mathcal{D}(h_T^*)}(h_T^*)$ is further dependent on the difference between the two classifiers' performances on the source data $\mathcal{D}_S$. This offers an opportunity to evaluate the possible error transferability using the source data only.

## 6 EXPERIMENTS

We perform synthetic experiments using real-world data to demonstrate our bounds. In particular, we use the FICO credit score data set (Board of Governors of the Federal Reserve System (US), 2007) which contains more than 300k records of TransUnion credit score of clients from different demographic groups. For our experiment on the preprocessed FICO data set (Hardt et al., 2016b), we convert the cumulative distribution function (CDF) of TransRisk score among different groups into group-wise credit score densities, from which we generate a balanced sample to represent a population where groups have equal representations. We demonstrate the application of our results in a series of resource allocations. We consider the hypothesis class of threshold classifiers and treat the classification outcome as the decision received by individuals.

For each time step $K = k$, we compute $h_S^*$, the statistical optimal classifier on the source distribution (i.e., the current reality for step $K = k$), and update the credit score for each individual according to the received decision as the new reality for time step $K = k + 1$. Details of the data generation is again deferred to Appendix C. We report our results in Figure 3. We do observe positive gaps $\text{Err}_{\mathcal{D}(h_S^*)}(h_S^*) - \text{Err}_{\mathcal{D}(h_T^*)}(h_T^*)$, indicating the suboptimality of training on $\mathcal{D}_S$. The gaps are well bounded by the theoretical upper bound (UB). Our lower bounds (LB) do return meaningful positive gaps, demonstrating the trade-offs that a classifier has to suffer on either the source distribution or the induced target distribution.

**Challenges in Minimizing Induced Risk and Concluding Remarks**  We presented a sequence of model transferability results for settings where agents will respond to a deployed model. The response leads to an induced distribution that the learner would not know before deploying the model. Our results cover for both a general response setting and for specific ones (covariate shift and target shift). Looking forward to solving the induced risk minimization, the literature of domain adaptation has provided us solutions to minimize the risk on the target distribution via a nicely developed set of results (Sugiyama et al., 2008; 2007; Shimodaira, 2000). This allows us to extend the solutions to minimize the induced risk too. Nonetheless we will highlight additional computational challenges. Let's use the covariate shift setting. The scenario for target shift is similar. For covariate shift, recall that earlier we derived the following fact:

$$\text{(Importance Reweighting)} : \mathbb{E}_{\mathcal{D}(h)}[\ell(h; X, Y)] = \mathbb{E}_{\mathcal{D}}[\omega_x(h) \cdot \ell(h; x, y)]. \tag{6}$$

This formula informs us that a promising solution that uses $\omega_x(h)$ to perform reweighted ERM. There are two primary challenges when carrying out optimization of the above objective. Of course, the primary challenge that stands in the way is how do we know $\omega_x(h)$. When one could build models to predict the response $\mathcal{D}(h)$ and then $\omega_x(h)$ (e.g., using the replicator dynamics model as we introduced earlier), one could rework the above loss and apply standard gradient descent approaches. We provide a concrete example and discussion in Appendix E. Without making any assumptions on the mapping between $h$ and $\mathcal{D}(h)$, one can only potentially rely on the *bandit feedbacks* from the decision subjects to estimate the influence of $h$ on $\mathcal{D}(h)$ - we also laid out a possibility in Appendix E too. It can also be inferred from Eqn. (6) that the second challenge is the induced risk minimization might not even be convex - due to the limit of space, we defer the detailed discussion again to the Appendix D.

## 7 ETHICAL STATEMENT

The primary goal of our study is to put human in the center when considering domain shift. The development of the paper is fully aware of any fairness concerns and we expect positive societal impact. Unawareness of the potential distribution shift might lead to unintended consequence when training a machine learning model. One goal of this paper is to raise awareness of this issue for a safe deployment of machine learning methods in high-stake societal applications.

A subset of our results are developed under assumptions (e.g., Theorem 4.6). Therefore we want to caution readers of potential misinterpretation of applicability of the reported theoretical guarantees. Our contributions are mostly theoretical and our experiments use synthetic agent models to simulate distribution shift. A future direction is to collect real human experiment data to support the findings. Our paper ends with discussing the challenges in learning under the responding distribution and other objectives that might arise.

We believe this is a promising research direction for the machine learning community, both as a unaddressed technical problem and a stepstone for putting human in the center when training a machine learning model.

## 8 REPRODUCIBILITY STATEMENT

We provide the following checklist for the purpose of reproducibility:

1. Generals:
   (a) Do the main claims made in the abstract and introduction accurately reflect the paper's contributions and scope? [Yes]
   (b) Did you describe the limitations of your work? [Yes] We have stated our assumptions and limitations of the results. We also discussed the limitations in the conclusion.
   (c) Did you discuss any potential negative societal impacts of your work? [Yes] One of our work's goals is to raise awareness of this issue for a safe deployment of machine learning methods in high-stake societal applications. We discuss the potential misinterpretation of our results in conclusion.
   (d) Have you read the ethics review guidelines and ensured that your paper conforms to them? [Yes]

2. If you are including theoretical results...
   (a) Did you state the full set of assumptions of all theoretical results? [Yes]
   (b) Did you include complete proofs of all theoretical results? [Yes] We present the complete proofs in the appendix.

3. If you ran experiments...
   (a) Did you include the code, data, and instructions needed to reproduce the main experimental results (either in the supplemental material or as a URL)? [Yes] We included experiment details in the appendix and submitted the implementation in the supplementary materials.
   (b) Did you specify all the training details (e.g., data splits, hyperparameters, how they were chosen)? [Yes]
   (c) Did you report error bars (e.g., with respect to the random seed after running experiments multiple times)? [N/A] In our controlled experiment, we do not tune parameters and do not observe a significant variance in the results.
   (d) Did you include the total amount of compute and the type of resources used (e.g., type of GPUs, internal cluster, or cloud provider)? [Yes]

4. If you are using existing assets (e.g., code, data, models) or curating/releasing new assets...
   (a) If your work uses existing assets, did you cite the creators? [Yes]
   (b) Did you mention the license of the assets? [Yes]
   (c) Did you include any new assets either in the supplemental material or as a URL? [No]

      (d) Did you discuss whether and how consent was obtained from people whose data you're using/curating? [N/A]

      (e) Did you discuss whether the data you are using/curating contains personally identifiable information or offensive content? [N/A]

5. If you used crowdsourcing or conducted research with human subjects...

      (a) Did you include the full text of instructions given to participants and screenshots, if applicable? [N/A]

      (b) Did you describe any potential participant risks, with links to Institutional Review Board (IRB) approvals, if applicable? [N/A]

      (c) Did you include the estimated hourly wage paid to participants and the total amount spent on participant compensation? [N/A]

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

# A  APPENDIX

We arrange the appendix as follows:

- Appendix A.1 provides some real life scenarios where transparent models are useful or required.
- Appendix A.2 provides comparisons of our setting and other sub-areas in domain adaptation.
- Appendix A.3 provides proof for Theorem 3.1.
- Appendix A.4 provides proof for Theorem 3.2.
- Appendix A.5 provides proof of Theorem 3.3.
- Appendix A.6 provides proof for Proposition 4.1.
- Appendix A.7 provides proof for Theorem 4.2.
- Appendix A.8 provides proof for Theorem 4.6.
- Appendix A.9 provides omitted assumptions and proof for Section 4.3.
- Appendix A.10 provides proof for Theorem 5.1.
- Appendix A.11 provides proof for Theorem B.1.
- Appendix A.12 provides proof for Proposition B.2.
- Appendix B provides additional lower bound and examples for the target shift setting.
- Appendix C provides missing experimental results , including new experimental results using synthetic datasets generated according to causal graphs defined in Figure 2. We also add additional experimental results on credit score data set.
- Appendix D discusses challenges in minimizing induced risk.
- Appendix E provides discussions on how to directly minimize the induced risk.
- Appendix F provides discussions on adding regularization to the objective function.
- Appendix G provides discussions on the tightness of our theoretical bounds.

## A.1  EXAMPLE USAGES OF TRANSPARENT MODELS

As we mentioned in Section 1, there is an increasing requirement of making the decision rule to be transparent due to its potential consequences impacts to individual decision subject. Here we provide the following reasons for using transparent models:

- Government regulation may require the model to be transparent, especially in public services;
- In some cases, companies may want to disclose their models so users will have explanations and are incentivized to better use the provided services.
- Regardless of whether models are published voluntarily, model parameters can often be inferred via well-known query "attacks".

In addition, we name some concrete examples of some real-life applications:

- Consider the *Medicaid health insurance program* in the United States, which serves low-income people. There is an obligation to provide transparency/disclose the rules (model to automate the decisions) that decide whether individuals qualify for the program — in fact, most public services have "terms" that are usually set in stone and explained in the documentation. Agents can observe the rules and will adapt their profiles to be qualified if needed. For instance, an agent can decide to provide additional documentation they need to guarantee approval. For more applications along these lines, please refer to this report[5].
- Credit score companies directly publish their criteria for assessing credit risk scores. In loan application settings, companies actually have the incentive to release criteria to incentivize agents to meet their qualifications and use their services.Furthermore, making decision models transparent will gain the trust of users.

---

[5]https://datasociety.net/library/poverty-lawgorithms/

- It is also known that it is possible to steal model parameters, if agents have incentives to do so[6]. For instance, spammers frequently infer detection mechanisms by sending different email variants; they then adjust their spam content accordingly.

## A.2 COMPARISON OF OUR SETTING AND SOME AREAS IN DOMAIN ADAPTATION

We compare our setting (We address it as IDA, representing "induced domain adaptation") with the following areas:

- Adversarial attack Chakraborty et al. (2018); Papernot et al. (2016); Song et al. (2019): in adversarial attack, the true label $Y$ stays the same for the attacked feature, while in IDA, we allow the true label to change as well. One can think of adversarial attack as a specific form of IDA where the induced distribution has a specific target, that is to maximize the classifier's error by only perturbing/modifying. Our transferability bound does, however, provide insights for how standard training results transfer to the attack setting.
- Domain generalization Wang et al. (2021b); Li et al. (2017); Muandet et al. (2013): the goal of domain generalization is to learn a model that can be generalized to any unseen distribution; Similar to our setting, one of the biggest challenges in domain generalization also the lack of target distribution during training. The major difference, however, is that our focus is to understand how the performance of a classifier trained on the source distribution degrades when evaluated on the induced distribution (which depends on how the population of decision subjects responds); this degradation depends on the classifier itself.
- Test-time adaptation Varsavsky et al. (2020); Wang et al. (2021a); Nado et al. (2021): the issue of test-time adaptation falls into the classical domain adaptation setting where the adaptation is independent of the model being deployed. Applying this technique to solve our problem requires accessing data (either unsupervised or supervised) drawn from $\mathcal{D}_S(h)$ for each $h$ being evaluated during different training epochs.

## A.3 PROOF OF THEOREM 3.1

*Proof.* We first establish two lemmas that will be helpful for bounding the errors of a pair of classifiers. Both are standard results from the domain adaption literature Ben-David et al. (2010).

**Lemma A.1.** *For any hypotheses $h, h' \in \mathcal{H}$ and distributions $\mathcal{D}, \mathcal{D}'$,*

$$|Err_{\mathcal{D}}(h, h') - Err_{\mathcal{D}'}(h, h')| \leq \frac{d_{\mathcal{H} \times \mathcal{H}}(\mathcal{D}, \mathcal{D}')}{2}.$$

*Proof.* Define the-cross prediction disagreement between two classifiers $h, h'$ on a distribution $\mathcal{D}$ as $\mathrm{Err}_{\mathcal{D}}(h, h') := \mathbb{P}_{\mathcal{D}}(h(X) \neq h'(X))$. By the definition of the $\mathcal{H}$−divergence,

$$\begin{aligned} d_{\mathcal{H} \times \mathcal{H}}(\mathcal{D}, \mathcal{D}') &= 2 \sup_{h, h' \in \mathcal{H}} |\mathbb{P}_{\mathcal{D}}(h(X) \neq h'(X)) - \mathbb{P}_{\mathcal{D}'}(h(X) \neq h'(X))| \\ &= 2 \sup_{h, h' \in \mathcal{H}} |\mathrm{Err}_{\mathcal{D}}(h, h') - \mathrm{Err}_{\mathcal{D}'}(h, h')| \\ &\geq 2 |\mathrm{Err}_{\mathcal{D}}(h, h') - \mathrm{Err}_{\mathcal{D}'}(h, h')| . \end{aligned}$$

$\square$

Another helpful lemma for us is the well-known fact that the 0-1 error obeys the triangle inequality (see, e.g., Crammer et al. (2008)):

**Lemma A.2.** *For any distribution $\mathcal{D}$ over instances and any labeling functions $f_1$, $f_2$, and $f_3$, we have $Err_{\mathcal{D}}(f_1, f_2) \leq Err_{\mathcal{D}}(f_1, f_3) + Err_{\mathcal{D}}(f_2, f_3)$.*

Denote by $\bar{h}^*$ the *ideal joint hypothesis*, which minimizes the combined error:

$$\bar{h}^* := \operatorname*{arg\,min}_{h' \in \mathcal{H}} \mathrm{Err}_{\mathcal{D}(h)}(h') + \mathrm{Err}_{\mathcal{D}_S}(h')$$

---

[6]https://www.wired.com/2016/09/how-to-steal-an-ai/

We have:

$$\begin{aligned}
\mathrm{Err}_{\mathcal{D}(h)}(h) &\leq \mathrm{Err}_{\mathcal{D}(h)}(\bar{h}^*) + \mathrm{Err}_{\mathcal{D}(h)}(h, \bar{h}^*) && \text{(Lemma A.2)} \\
&\leq \mathrm{Err}_{\mathcal{D}(h)}(\bar{h}^*) + \mathrm{Err}_{\mathcal{D}_S}(h, \bar{h}^*) + \left| \mathrm{Err}_{\mathcal{D}(h)}(h, \bar{h}^*) - \mathrm{Err}_{\mathcal{D}_S}(h, \bar{h}^*) \right| \\
&\leq \mathrm{Err}_{\mathcal{D}(h)}(\bar{h}^*) + \mathrm{Err}_{\mathcal{D}_S}(h) + \mathrm{Err}_{\mathcal{D}_S}(\bar{h}^*) + \frac{1}{2} d_{\mathcal{H} \times \mathcal{H}}(\mathcal{D}_S, \mathcal{D}(h)) && \text{(Lemma A.1)} \\
&= \mathrm{Err}_{\mathcal{D}_S}(h) + \lambda_{\mathcal{D}_S \to \mathcal{D}(h)} + \frac{1}{2} d_{\mathcal{H} \times \mathcal{H}}(\mathcal{D}_S, \mathcal{D}(h)). && \text{(Definition of } \bar{h}^*)
\end{aligned}$$

$\square$

## A.4 PROOF OF THEOREM 3.2

*Proof.* Invoking Theorem 3.1, and replacing $h$ with $h_T^*$ and $S$ with $\mathcal{D}(h_T^*)$, we have

$$\mathrm{Err}_{\mathcal{D}(h)}(h_T^*) \leq \mathrm{Err}_{\mathcal{D}(h_T^*)}(h_T^*) + \lambda_{\mathcal{D}(h) \to \mathcal{D}(h_T^*)} + \frac{1}{2} d_{\mathcal{H} \times \mathcal{H}}(\mathcal{D}(h_T^*), \mathcal{D}(h)) \tag{7}$$

Now observe that

$$\begin{aligned}
\mathrm{Err}_{\mathcal{D}(h)}(h) &\leq \mathrm{Err}_{\mathcal{D}(h)}(h_T^*) + \mathrm{Err}_{\mathcal{D}(h)}(h, h_T^*) \\
&\leq \mathrm{Err}_{\mathcal{D}(h)}(h_T^*) + \mathrm{Err}_{\mathcal{D}(h_T^*)}(h, h_T^*) + \left| \mathrm{Err}_{\mathcal{D}(h)}(h, h_T^*) - \mathrm{Err}_{\mathcal{D}(h_T^*)}(h, h_T^*) \right| \\
&\leq \mathrm{Err}_{\mathcal{D}(h)}(h_T^*) + \mathrm{Err}_{\mathcal{D}(h_T^*)}(h, h_T^*) + \frac{1}{2} d_{\mathcal{H} \times \mathcal{H}}(\mathcal{D}(h_T^*), \mathcal{D}(h)) && \text{(by Lemma A.1)} \\
&\leq \mathrm{Err}_{\mathcal{D}(h)}(h_T^*) + \mathrm{Err}_{\mathcal{D}(h_T^*)}(h) + \mathrm{Err}_{\mathcal{D}(h_T^*)}(h_T^*) + \frac{1}{2} d_{\mathcal{H} \times \mathcal{H}}(\mathcal{D}(h_T^*), \mathcal{D}(h)) \\
& && \text{(by Lemma A.2)} \\
&\leq \mathrm{Err}_{\mathcal{D}(h_T^*)}(h_T^*) + \lambda_{\mathcal{D}(h) \to \mathcal{D}(h_T^*)} + \frac{1}{2} d_{\mathcal{H} \times \mathcal{H}}(\mathcal{D}(h_T^*), \mathcal{D}(h)) && \text{(by equation 7)} \\
&\quad + \mathrm{Err}_{\mathcal{D}(h_T^*)}(h) + \mathrm{Err}_{\mathcal{D}(h_T^*)}(h_T^*) + \frac{1}{2} d_{\mathcal{H} \times \mathcal{H}}(\mathcal{D}(h_T^*), \mathcal{D}(h))
\end{aligned}$$

Adding $\mathrm{Err}_{\mathcal{D}(h)}(h)$ to both sides and rearranging terms yields

$$\begin{aligned}
2\mathrm{Err}_{\mathcal{D}(h)}(h) - 2\mathrm{Err}_{\mathcal{D}(h_T^*)}(h_T^*) &\leq \mathrm{Err}_{\mathcal{D}(h)}(h) + \mathrm{Err}_{\mathcal{D}(h_T^*)}(h) + \lambda_{\mathcal{D}(h) \to \mathcal{D}(h_T^*)} + d_{\mathcal{H} \times \mathcal{H}}(\mathcal{D}(h_T^*), \mathcal{D}(h)) \\
&= \Lambda_{\mathcal{D}(h) \to \mathcal{D}(h_T^*)}(h) + \lambda_{\mathcal{D}(h) \to \mathcal{D}(h_T^*)} + d_{\mathcal{H} \times \mathcal{H}}(\mathcal{D}(h_T^*), \mathcal{D}(h))
\end{aligned}$$

Dividing both sides by 2 completes the proof. $\square$

## A.5 PROOF OF THEOREM 3.3

*Proof.* Using the triangle inequality of $d_{\mathrm{TV}}$, we have

$$d_{\mathrm{TV}}(\mathcal{D}_{Y|S}, \mathcal{D}_Y(h)) \leq d_{\mathrm{TV}}(\mathcal{D}_{Y|S}, \mathcal{D}_{h|S}) + d_{\mathrm{TV}}(\mathcal{D}_{h|S}, \mathcal{D}_h(h)) + d_{\mathrm{TV}}(\mathcal{D}_h(h), \mathcal{D}_Y(h)) \tag{8}$$

and by the definition of $d_{\mathrm{TV}}$, the divergence term $d_{\mathrm{TV}}(\mathcal{D}_{Y|S}, \mathcal{D}_Y(h))$ becomes

$$\begin{aligned}
d_{\mathrm{TV}}(\mathcal{D}_{Y|S}, \mathcal{D}_{h|S}) &= |\mathbb{P}_{\mathcal{D}_S}(Y = +1) - \mathbb{P}_{\mathcal{D}_S}(h(x) = +1)| \\
&= \left| \frac{\mathbb{E}_{\mathcal{D}_S}[Y] + 1}{2} - \frac{\mathbb{E}_{\mathcal{D}_S}[h(X)] + 1}{2} \right| \\
&= \left| \frac{\mathbb{E}_{\mathcal{D}_S}[Y]}{2} - \frac{\mathbb{E}_{\mathcal{D}_S}[h(X)]}{2} \right| \\
&\leq \frac{1}{2} \cdot \mathbb{E}_{\mathcal{D}_S}[|Y - h(X)|] \\
&= \mathrm{Err}_{\mathcal{D}_S}(h)
\end{aligned}$$

Similarly, we have

$$d_{\mathrm{TV}}(\mathcal{D}_h(h), \mathcal{D}_Y(h)) \leq \mathrm{Err}_{\mathcal{D}(h)}(h)$$

As a result, we have

$$\text{Err}_{\mathcal{D}_S}(h) + \text{Err}_{\mathcal{D}(h)}(h) \geq d_{\text{TV}}(\mathcal{D}_{Y|S}, \mathcal{D}_{h|S}) + d_{\text{TV}}(\mathcal{D}_h(h), \mathcal{D}_Y(h))$$
$$\geq d_{\text{TV}}(\mathcal{D}_{Y|S}, \mathcal{D}_Y(h)) - d_{\text{TV}}(\mathcal{D}_{h|S}, \mathcal{D}_h(h)) \qquad \text{(by equation 8)}$$

which implies

$$\max\{\text{Err}_{\mathcal{D}_S}(h), \text{Err}_{\mathcal{D}(h)}(h)\} \geq \frac{d_{\text{TV}}(\mathcal{D}_{Y|S}, \mathcal{D}_Y(h)) - d_{\text{TV}}(\mathcal{D}_{h|S}, \mathcal{D}_h(h))}{2} .$$

$\square$

## A.6 PROOF OF PROPOSITION 4.1

*Proof.*

$$\mathbb{E}_{\mathcal{D}(h)}[\ell(h; X, Y)]$$
$$= \int \mathbb{P}_{\mathcal{D}(h)}(X = x, Y = y)\ell(h; x, y) \, dxdy$$
$$= \int \mathbb{P}_{\mathcal{D}_S}(Y = y|X = x) \cdot \mathbb{P}_{\mathcal{D}(h)}(X = x)\ell(h; x, y) \, dxdy$$
$$= \int \mathbb{P}_{\mathcal{D}_S}(Y = y|X = x) \cdot \mathbb{P}_{\mathcal{D}_S}(X = x) \cdot \frac{\mathbb{P}_{\mathcal{D}(h)}(X = x)}{\mathbb{P}_{\mathcal{D}_S}(X = x)} \cdot \ell(h; x, y) \, dxdy$$
$$= \int \mathbb{P}_{\mathcal{D}_S}(Y = y|X = x) \cdot \mathbb{P}_{\mathcal{D}_S}(X = x) \cdot \omega_x(h) \cdot \ell(h; x, y) \, dxdy$$
$$= \mathbb{E}_{\mathcal{D}_S}[\omega_x(h) \cdot \ell(h; x, y)]$$

$\square$

## A.7 PROOF OF THEOREM 4.2

*Proof.* We start from the error induced by $h_S^*$. Let the *average importance weight induced by $h_S^*$* be $\bar{\omega}(h_S^*) = \mathbb{E}_{\mathcal{D}_S}[\omega_x(h_S^*)]$; we add and subtract this from the error:

$$\text{Err}_{\mathcal{D}(h_S^*)}(h_S^*) = \mathbb{E}_{\mathcal{D}_S}\left[\omega_x(h_S^*) \cdot \mathbb{1}(h_S^*(x) \neq y)\right]$$
$$= \mathbb{E}_{\mathcal{D}_S}\left[\bar{\omega}(h_S^*) \cdot \mathbb{1}(h_S^*(x) \neq y)\right] + \mathbb{E}_{\mathcal{D}_S}\left[(\omega_x(h_S^*) - \bar{\omega}(h_S^*)) \cdot \mathbb{1}(h_S^*(x) \neq y)\right]$$

In fact, $\bar{\omega}(h_S^*) = 1$, since

$$\bar{\omega}(h_S^*) = \mathbb{E}_{\mathcal{D}_S}[\omega_x(h_S^*)] = \int \omega_x(h_S^*)\mathbb{P}_{\mathcal{D}_S}(X = x)dx$$
$$= \int \frac{\mathbb{P}_{\mathcal{D}(h)}(X = x)}{\mathbb{P}_{\mathcal{D}_S}(X = x)}\mathbb{P}_{\mathcal{D}_S}(X = x)dx = \int \mathbb{P}_{\mathcal{D}(h)}(X = x)dx = 1$$

Now consider any other classifier $h$. We have

$$\text{Err}_{\mathcal{D}(h_S^*)}(h_S^*)$$
$$= \mathbb{E}_{\mathcal{D}_S}\left[\mathbb{1}(h_S^*(x) \neq y)\right] + \mathbb{E}_{\mathcal{D}_S}\left[(\omega_x(h_S^*) - \bar{\omega}(h_S^*)) \cdot \mathbb{1}(h_S^*(x) \neq y)\right]$$
$$\leq \mathbb{E}_{\mathcal{D}_S}\left[\mathbb{1}(h(x) \neq y)\right] + \mathbb{E}_{\mathcal{D}_S}\left[(\omega_x(h_S^*) - \bar{\omega}(h_S^*)) \cdot \mathbb{1}(h_S^*(x) \neq y)\right]$$
$$\qquad\qquad\qquad\qquad\qquad\qquad\qquad \text{(by optimality of } h_S^* \text{ on } \mathcal{D}_S)$$
$$= \mathbb{E}_{\mathcal{D}_S}\left[\bar{\omega}(h) \cdot \mathbb{1}(h(x) \neq y)\right] + \mathbb{E}_{\mathcal{D}_S}\left[(\omega_x(h_S^*) - \bar{\omega}(h_S^*)) \cdot \mathbb{1}(h_S^*(x) \neq y)\right]$$
$$\qquad\qquad\qquad\qquad\qquad\qquad\qquad \text{(multiply by } \bar{\omega}(h_S^*) = 1)$$
$$= \mathbb{E}_{\mathcal{D}_S}\left[\omega_x(h) \cdot \mathbb{1}(h(x) \neq y)\right] + \mathbb{E}_{\mathcal{D}_S}\left[(\bar{\omega}(h) - \omega_x(h)) \cdot \mathbb{1}(h(x) \neq y)\right]$$
$$\qquad\qquad\qquad\qquad\qquad\qquad\qquad \text{(add and subtract } \bar{\omega}(h_S^*))$$
$$\qquad + \mathbb{E}_{\mathcal{D}_S}\left[(\omega_x(h_S^*) - \bar{\omega}(h_S^*)) \cdot \mathbb{1}(h_S^*(x) \neq y)\right]$$
$$= \text{Err}_{\mathcal{D}(h)}(h) + \text{Cov}(\omega_x(h_S^*), \mathbb{1}(h_S^*(x) \neq y)) - \text{Cov}(\omega_x(h), \mathbb{1}(h(x) \neq y))$$

Moving the error terms to one side, we have

$$
\begin{aligned}
&\mathrm{Err}_{\mathcal{D}(h_S^*)}(h_S^*) - \mathrm{Err}_{\mathcal{D}(h)}(h) \\
&\leq \mathrm{Cov}(\omega_x(h_S^*), \mathbb{1}(h_S^*(x) \neq y)) - \mathrm{Cov}(\omega_x(h), \mathbb{1}(h(x) \neq y)) \\
&\leq \sqrt{\mathrm{Var}(\omega_x(h_S^*)) \cdot \mathrm{Var}(\mathbb{1}(h_S^*(x) \neq y))} \qquad\qquad (|\mathrm{Cov}(X,Y)| \leq \sqrt{\mathrm{Var}(X) \cdot \mathrm{Var}(Y)}) \\
&\quad + \sqrt{\mathrm{Var}(\omega_x(h)) \cdot \mathrm{Var}(\mathbb{1}(h(x) \neq y))} \\
&= \sqrt{\mathrm{Var}(\omega_x(h_S^*)) \cdot \mathrm{Err}_S(h_S^*)(1 - \mathrm{Err}_S(h_S^*))} + \sqrt{\mathrm{Var}(\omega_x(h)) \cdot \mathrm{Err}_{\mathcal{D}_S}(h)(1 - \mathrm{Err}_{\mathcal{D}_S}(h))} \\
&\leq \sqrt{\mathrm{Var}(\omega_x(h_S^*)) \cdot \mathrm{Err}_S(h_S^*)} + \sqrt{\mathrm{Var}(\omega_x(h)) \cdot \mathrm{Err}_{\mathcal{D}_S}(h)} \qquad\qquad (1 - \mathrm{Err}_{\mathcal{D}_S}(h) \leq 1) \\
&\leq \sqrt{\mathrm{Err}_{\mathcal{D}_S}(h)} \cdot \left( \sqrt{\mathrm{Var}(\omega_x(h_S^*))} + \sqrt{\mathrm{Var}(\omega_x(h))} \right)
\end{aligned}
$$

Since this holds for any $h$, it certainly holds for $h = h_T^*$.

$\square$

### A.8 OMITTED ASSUMPTIONS AND PROOF OF THEOREM 4.6

Denote $X_+(h) = \{x : \omega_x(h) \geq 1\}$ and $X_-(h) = \{x : \omega_x(h) < 1\}$. First we observe that

$$
\begin{aligned}
&\int_{X_+(h)} \mathbb{P}_{\mathcal{D}_S}(X = x)(1 - \omega_x(h))dx \\
&+ \int_{X_-(h)} \mathbb{P}_{\mathcal{D}_S}(X = x)(1 - \omega_x(h))dx = 0
\end{aligned}
$$

This is simply because of $\int_x \mathbb{P}_{\mathcal{D}_S}(X = x) \cdot \omega_x(h)dx = \int_x \mathbb{P}_{\mathcal{D}(h)}(X = x)dx = 1$.

*Proof.* Notice that in the setting of binary classification, we can write the total variation distance between $\mathcal{D}_{Y|S}$ and $\mathcal{D}_Y(h)$ as the difference between the probability of $Y = +1$ and the probability of $Y = -1$:

$$
\begin{aligned}
&d_{\mathrm{TV}}(\mathcal{D}_{Y|S}, \mathcal{D}_Y(h)) \\
&= \left| \mathbb{P}_{\mathcal{D}_S}(Y = +1) - \mathbb{P}_{\mathcal{D}(h)}(Y = +1) \right| \\
&= \left| \int \mathbb{P}_{\mathcal{D}_S}(Y = +1|X = x)\mathbb{P}_{\mathcal{D}_S}(X = x)dx - \int \mathbb{P}_{\mathcal{D}_S}(Y = +1|X = x)\mathbb{P}_{\mathcal{D}_S}(X = x)\omega_x(h)dx \right| \\
&= \left| \int \mathbb{P}_{\mathcal{D}_S}(Y = +1|X = x)\mathbb{P}_{\mathcal{D}_S}(X = x) \cdot (1 - \omega_x(h))dx \right| \qquad (9)
\end{aligned}
$$

Similarly we have

$$
d_{\mathrm{TV}}(\mathcal{D}_{h|S}, \mathcal{D}_h(h)) = \left| \int \mathbb{P}_{\mathcal{D}_S}(h(x) = +1|X = x)\mathbb{P}_{\mathcal{D}_S}(X = x) \cdot (1 - \omega_x(h))dx \right| \qquad (10)
$$

We can further expand the total variation distance between $\mathcal{D}_{Y|S}$ and $\mathcal{D}_Y(h)$ as follows:

$$d_{\text{TV}}(\mathcal{D}_{Y|S}, \mathcal{D}_Y(h))$$

$$= \left| \int \mathbb{P}_{\mathcal{D}_S}(Y = +1 | X = x) \mathbb{P}_{\mathcal{D}_S}(X = x) \cdot (1 - \omega_x(h)) dx \right|$$

$$= \Big| \underbrace{\int_{X_+(h)} \mathbb{P}_{\mathcal{D}}(Y = +1 | X = x) \mathbb{P}_{\mathcal{D}_S}(X = x) \cdot (1 - \omega_x(h)) dx}_{\leq 0}$$

$$+ \underbrace{\int_{X_-(h)} \mathbb{P}_{\mathcal{D}_S}(Y = +1 | X = x) \mathbb{P}_{\mathcal{D}_S}(X = x) \cdot (1 - \omega_x(h)) dx \Big|}_{> 0}$$

$$= - \int_{X_+(h)} \mathbb{P}_{\mathcal{D}_S}(Y = +1 | X = x) \mathbb{P}_{\mathcal{D}_S}(X = x) \cdot (1 - \omega_x(h)) dx$$

$$- \int_{X_-(h)} \mathbb{P}_{\mathcal{D}_S}(Y = +1 | X = x) \mathbb{P}_{\mathcal{D}_S}(X = x) \cdot (1 - \omega_x(h)) dx \qquad \text{(by Assumption 4.3)}$$

$$= \int_{X_+(h)} \mathbb{P}_{\mathcal{D}_S}(Y = +1 | X = x) \mathbb{P}_{\mathcal{D}_S}(X = x) \cdot (\omega_x(h) - 1) dx$$

$$+ \int_{X_-(h)} \mathbb{P}_{\mathcal{D}_S}(Y = +1 | X = x) \mathbb{P}_{\mathcal{D}_S}(X = x) \cdot (\omega_x(h) - 1) dx \qquad \text{(by equation 9)}$$

$$= \int \mathbb{P}_{\mathcal{D}_S}(Y = +1 | X = x) \mathbb{P}_{\mathcal{D}_S}(X = x) \cdot (\omega_x(h) - 1) dx$$

Similarly, by assumption 4.4 and equation equation 10, we have

$$d_{\text{TV}}(\mathcal{D}_{h|S}, \mathcal{D}_h(h)) = \int \mathbb{P}_{\mathcal{D}_S}(h(x) = +1 | X = x) \mathbb{P}_{\mathcal{D}_S}(X = x) \cdot (\omega_x(h) - 1) dx$$

Thus we can bound the difference between $d_{\text{TV}}(\mathcal{D}_{Y|S}, \mathcal{D}_Y(h))$ and $d_{\text{TV}}(\mathcal{D}_{h|S}, \mathcal{D}_h(h))$ as follows:

$$d_{\text{TV}}(\mathcal{D}_{Y|S}, \mathcal{D}_Y(h)) - d_{\text{TV}}(\mathcal{D}_{h|S}, \mathcal{D}_h(h))$$

$$= \int \mathbb{P}_{\mathcal{D}_S}(Y = +1 | X = x) \mathbb{P}_{\mathcal{D}_S}(X = x) \cdot (\omega_x(h) - 1) dx$$

$$- \int \mathbb{P}_{\mathcal{D}}(h(x) = +1 | X = x) \mathbb{P}_{\mathcal{D}_S}(X = x) \cdot (\omega_x(h) - 1) dx$$

$$= \int [\mathbb{P}_{\mathcal{D}_S}(Y = +1 | X = x) - \mathbb{P}_{\mathcal{D}_S}(h(x) = +1 | X = x)] \mathbb{P}_{\mathcal{D}_S}(X = x) \cdot (\omega_x(h) - 1) dx$$

$$= \mathbb{E}_{\mathcal{D}_S}[(\mathbb{P}_{\mathcal{D}_S}(Y = +1 | X = x) - \mathbb{P}_{\mathcal{D}_S}(h(x) = +1 | X = x)) (\omega_x(h) - 1)]$$
$$\text{(by Assumption 4.5)}$$

$$> \mathbb{E}_{\mathcal{D}_S}[\mathbb{P}_{\mathcal{D}_S}(Y = +1 | X = x) - \mathbb{P}_{\mathcal{D}_S}(h(x) = +1 | X = x)] \mathbb{E}_{\mathcal{D}_S}[\omega_x(h) - 1]$$
$$= 0$$

Combining the above with Theorem 3.3, we have

$$\max\{\text{Err}_{\mathcal{D}_S}(h), \text{Err}_{\mathcal{D}(h)}(h)\} \geq \frac{d_{\text{TV}}(\mathcal{D}_{Y|S}, \mathcal{D}_Y(h)) - d_{\text{TV}}(\mathcal{D}_{h|S}, \mathcal{D}_h(h))}{2} > 0$$

$$\square$$

## A.9 OMITTED DETAILS FOR SECTION 4.3

With Setup 2 - Setup 4, we can further specify the important weight $w_x(h)$ for the strategic response setting:

**Lemma A.3.** *Recall the definition for the covariate shift important weight coefficient* $\omega_x(h) := \frac{\mathbb{P}_{D(h)}(X=x)}{\mathbb{P}_{D_S}(X=x)}$, *for our strategic response setting, we have,*

$$
w_x(h) = \begin{cases} 1, & x \in [0, \tau_h - B) \\ \frac{\tau_h - x}{B}, & x \in [\tau_h - B, \tau_h) \\ \frac{1}{B}(-x + \tau_h + 2B), & x \in [\tau_h, \tau_h + B) \\ 1, & x \in [\tau_h + B, 1] \end{cases} \tag{11}
$$

Proof for Lemma A.3:

*Proof.* We discuss the induced distribution $\mathcal{D}(h)$ by cases:

- For the features distributed between $[0, \tau_h - B]$: since we assume the agents are rational, under assumption 2, agents with feature that is smaller than $[0, \tau_h - B]$ will not perform any kinds of adaptations, and no other agents will adapt their features to this range of features either, so the distribution between $[0, \tau_h - B]$ will remain the same as before.

- For the target distribution between $[\tau_h - B, \tau_h]$ can be directly calculated from assumption 3.

- For distribution between $[\tau_h, \tau_h + B]$, consider a particular feature $x^\star \in [\tau_h, \tau_h + B]$, under Setup 4, we know its new distribution becomes:

$$
\begin{aligned}
\mathbb{P}_{\mathcal{D}(h)}(x = x^\star) &= 1 + \int_{x^\star - B}^{\tau_h} \frac{1 - \frac{\tau_h - z}{B}}{B - \tau_h + z} dz \\
&= 1 + \int_{x^\star - B}^{\tau_h} \frac{1}{B} dz \\
&= \frac{1}{B}(-x^\star + \tau_h + 2B)
\end{aligned}
$$

- For the target distribution between $[\tau_h + B, 1]$: under assumption 2 and 4, we know that no agents will change their feature to this feature region. So the distribution between $[\tau_h + B, 1]$ remains the same as the source distribution.

Recall the definition for the covariate shift important weight coefficient $\omega_x(h) := \frac{\mathbb{P}_{D(h)}(X=x)}{\mathbb{P}_{D_S}(X=x)}$, the distribution of $\omega_x(h)$ after agents' strategic responding becomes:

$$
\omega_x(h) = \begin{cases} 1, & x \in [0, \tau_h - B) \text{ and } x \in [\tau_h + B, 1] \\ \frac{\tau_h - x}{B}, & x \in [\tau_h - B, \tau_h) \\ \frac{1}{B}(-x + \tau_h + 2B), & x \in [\tau_h, \tau_h + B) \\ 0, & \text{otherwise} \end{cases} \tag{12}
$$

$\square$

Proof for Proposition 4.7:

*Proof.* According to Lemma A.3, we can compute the variance of $w_x(h)$ as $\mathrm{Var}(w_x(h)) = \mathbb{E}(w_x(h)^2) - \mathbb{E}(w_x(h)^2) = \frac{2}{3}B$. Then by plugging it to the general bound for Theorem 4.2 gives us the desirable result. $\square$

### A.10 PROOF OF THEOREM 5.1

*Proof.* Defining $p := \mathbb{P}_{\mathcal{D}_S}(Y = +1)$, $p(h) = \mathbb{P}_{\mathcal{D}(h)}(Y = +1)$, we have

$$\text{Err}_{\mathcal{D}(h_S^*)}(h_S^*) = p(h_S^*) \cdot \text{Err}_+(h_S^*) + (1 - p(h_S^*)) \cdot \text{Err}_-(h_S^*)$$

$$\text{(by definitions of } p(h_S^*), \text{Err}_+(h_S^*), \text{ and Err}_-(h_S^*))$$

$$= \underbrace{p \cdot \text{Err}_+(h_S^*) + (1 - p) \cdot \text{Err}_-(h_S^*)}_{\text{(I)}} + (p(h_S^*) - p)[\text{Err}_+(h_S^*) - \text{Err}_-(h_S^*)] \quad (13)$$

We can expand (I) as follows:

$$p \cdot \text{Err}_+(h_S^*) + (1 - p) \cdot \text{Err}_-(h_S^*)$$
$$\leq p \cdot \text{Err}_+(h_T^*) + (1 - p) \cdot \text{Err}_-(h_T^*) \qquad \text{(by optimality of } h_S^* \text{ on } \mathcal{D}_S)$$
$$= p(h_T^*) \cdot \text{Err}_+(h_T^*) + (1 - p(h_T^*)) \cdot \text{Err}_-(h_T^*) + (p - p(h_T^*)) \cdot [\text{Err}_+(h_T^*) - \text{Err}_-(h_T^*)]$$
$$= \text{Err}_{\mathcal{D}(h_T^*)}(h_T^*) + (p - p(h_T^*)) \cdot [\text{Err}_+(h_T^*) - \text{Err}_-(h_T^*)] .$$

Plugging this back into equation 13, we have

$$\text{Err}_{\mathcal{D}(h_S^*)}(h_S^*) - \text{Err}_{\mathcal{D}(h_T^*)}(h_T^*) \leq (p(h_S^*) - p)[\text{Err}_+(h_S^*) - \text{Err}_-(h_S^*)] + (p - p(h_T^*)) \cdot [\text{Err}_+(h_T^*) - \text{Err}_-(h_T^*)]$$

Notice that

$$0.5(\text{Err}_+(h) - \text{Err}_-(h)) = 0.5 \cdot 1 - 0.5 \cdot \mathbb{P}(h(X) = +1 | Y = +1) - 0.5 \cdot \mathbb{P}(h(X) = +1 | Y = -1)$$
$$= 0.5 - \mathbb{P}_{\mathcal{D}_u}(h(X) = +1)$$

where $\mathcal{D}_u$ is a distribution with uniform prior. Then

$$(p(h_S^*) - p)[\text{Err}_+(h_S^*) - \text{Err}_-(h_S^*)] = 2(p(h_S^*) - p) \cdot (0.5 - \mathbb{P}_{\mathcal{D}_u}(h(X) = +1))$$
$$(p - p(h_T^*))[\text{Err}_+(h_T^*) - \text{Err}_-(h_T^*)] = 2(p - p(h_T^*)) \cdot (0.5 - \mathbb{P}_{\mathcal{D}_u}(h(X) = +1))$$

Adding together these two equations yields

$$(p(h_S^*) - p)[\text{Err}_+(h_S^*) - \text{Err}_-(h_S^*)] + (p - p(h_T^*)) \cdot [\text{Err}_+(h_T^*) - \text{Err}_-(h_T^*)]$$
$$= 2(p(h_S^*) - p) \cdot (0.5 - \mathbb{P}_{\mathcal{D}_u}(h_S^*(X) = +1)) + 2(p - p(h_T^*)) \cdot (0.5 - \mathbb{P}_{\mathcal{D}_u}(h_T^*(X) = +1))$$
$$= (p(h_S^*) - p(h_T^*)) - 2 (p(h_S^*)\mathbb{P}_{\mathcal{D}_u}(h_S^*(X) = +1) - p(h_T^*)\mathbb{P}_{\mathcal{D}_u}(h_T^*(X) = +1))$$
$$\quad + 2p \cdot (\mathbb{P}_{\mathcal{D}_u}(h_S^*(X) = +1) - \mathbb{P}_{\mathcal{D}_u}(h_T^*(X) = +1))$$
$$\leq |p(h_S^*) - p(h_T^*)| \cdot (1 + 2|\mathbb{P}_{\mathcal{D}_u}(h_S^*(X) = +1) - \mathbb{P}_{\mathcal{D}_u}(h_T^*(X) = +1)|)$$
$$\quad + 2p \cdot |\mathbb{P}_{\mathcal{D}_u}(h_S^*(X) = +1) - \mathbb{P}_{\mathcal{D}_u}(h_T^*(X) = +1)| \quad (14)$$

Meanwhile,

$$|\mathbb{P}_{\mathcal{D}_u}(h_S^*(X) = +1) - \mathbb{P}_{\mathcal{D}_u}(h_T^*(X) = +1)|$$
$$\leq 0.5 \cdot |\mathbb{P}_{\mathcal{D}|Y=+1}(h_S^*(X) = +1) - \mathbb{P}_{\mathcal{D}|Y=+1}(h_T^*(X) = +1)|$$
$$\quad + 0.5 \cdot |\mathbb{P}_{\mathcal{D}|Y=-1}(h_S^*(X) = +1) - \mathbb{P}_{\mathcal{D}|Y=-1}(h_T^*(X) = +1)|$$
$$= 0.5 (d_{\text{TV}}(\mathcal{D}_+(h_S^*), \mathcal{D}_+(h_T^*)) + d_{\text{TV}}(\mathcal{D}_-(h_S^*), \mathcal{D}_-(h_T^*))) \quad (15)$$

Combining equation 14 and equation 15 gives

$$|p(h_S^*) - p(h_T^*)| \cdot (1 + 2 \cdot |\mathbb{P}_{\mathcal{D}_u}(h_S^*(X) = +1) - \mathbb{P}_{\mathcal{D}_u}(h_T^*(X) = +1)|)$$
$$\quad + 2p \cdot |\mathbb{P}_{\mathcal{D}_u}(h_S^*(X) = +1) - \mathbb{P}_{\mathcal{D}_u}(h_T^*(X) = +1)|$$
$$\leq |p(h_S^*) - p(h_T^*)| \cdot (1 + d_{\text{TV}}(\mathcal{D}_+(h_S^*), \mathcal{D}_+(h_T^*)) + d_{\text{TV}}(\mathcal{D}_-(h_S^*), \mathcal{D}_-(h_T^*))$$
$$\quad + p \cdot (d_{\text{TV}}(\mathcal{D}_+(h_S^*), \mathcal{D}_+(h_T^*)) + d_{\text{TV}}(\mathcal{D}_-(h_S^*), \mathcal{D}_-(h_T^*)))$$
$$\leq |p(h_S^*) - p(h_T^*)| + (1 + p) \cdot (d_{\text{TV}}(\mathcal{D}_+(h_S^*), \mathcal{D}_+(h_T^*)) + d_{\text{TV}}(\mathcal{D}_-(h_S^*), \mathcal{D}_-(h_T^*))) .$$

$$\square$$

## A.11    PROOF OF THEOREM B.1

We will make use of the following fact:

**Lemma A.4.** *Under label shift, $TPR_S(h) = TPR_h(h)$ and $FPR_S(h) = FPR_h(h)$.*

*Proof.* We have

$$
\begin{aligned}
\mathrm{TPR}_h(h) =& \mathbb{P}_{\mathcal{D}(h)}(h(X) = +1|Y = +1)\\
=& \int \mathbb{P}_{\mathcal{D}(h)}(h(X) = +1, X = x|Y = +1)dx\\
=& \int \mathbb{P}_{\mathcal{D}(h)}(h(X) = +1|X = x, Y = +1)\mathbb{P}_{\mathcal{D}(h)}(X = x|Y = +1)dx\\
=& \int \mathbb{1}(h(x) = +1)\mathbb{P}_{\mathcal{D}(h)}(X = x|Y = +1)dx\\
=& \int \mathbb{1}(h(x) = +1)\mathbb{P}_{\mathcal{D}_S}(X = x|Y = +1)dx \qquad \text{(by definition of label shift)}\\
=& \int \mathbb{P}_{\mathcal{D}_S}(h(X) = +1|X = x, Y = +1)\mathbb{P}_{\mathcal{D}_S}(X = x|Y = +1)dx\\
=& \mathrm{TPR}_S(h)
\end{aligned}
$$

The argument for $\mathrm{TPR}_h(h) = \mathrm{TPR}_S(h)$ is analogous. $\qquad\square$

Now we proceed to prove the theorem.

*Proof of Theorem B.1.* In section 3.2 we showed a general lower bound on the maximum of $\mathrm{Err}_{\mathcal{D}_S}(h)$ and $\mathrm{Err}_{\mathcal{D}(h)}(h)$:

$$
\max\{\mathrm{Err}_{\mathcal{D}_S}(h), \mathrm{Err}_{\mathcal{D}(h)}(h)\} \geq \frac{d_{\mathrm{TV}}(\mathcal{D}_{Y|S}, \mathcal{D}_Y(h)) - d_{\mathrm{TV}}(\mathcal{D}_{h|S}, \mathcal{D}_h(h))}{2}
$$

In the case of label shift, and by the definitions of $p$ and $p(h)$,

$$
d_{\mathrm{TV}}(\mathcal{D}_{Y|S}, \mathcal{D}_Y(h)) = |\mathbb{P}_{\mathcal{D}_S}(Y = +1) - \mathbb{P}_{\mathcal{D}(h)}(Y = +1)| = |p - p(h)| \tag{16}
$$

In addition, we have

$$
\mathcal{D}_{h|S} = \mathbb{P}_S(h(X) = +1) = p \cdot \mathrm{TPR}_S(h) + (1 - p) \cdot \mathrm{FPR}_S(h) \tag{17}
$$

Similarly

$$
\begin{aligned}
\mathcal{D}_h(h) =& \mathbb{P}_{\mathcal{D}(h)}(h(X) = +1)\\
=& p(h) \cdot \mathrm{TPR}_h(h) + (1 - p(h)) \cdot \mathrm{FPR}_h(h)\\
=& p(h) \cdot \mathrm{TPR}_S(h) + (1 - p(h)) \cdot \mathrm{FPR}_S(h) \qquad \text{(by Lemma A.4)} \tag{18}
\end{aligned}
$$

Therefore

$$
\begin{aligned}
d_{\mathrm{TV}}(\mathcal{D}_{h|S}, \mathcal{D}_h(h)) =& |\mathbb{P}_{\mathcal{D}_S}(h(X) = +1) - \mathbb{P}_{\mathcal{D}(h)}(h(X) = +1)|\\
=& |(p - p(h)) \cdot \mathrm{TPR}_S(h) + (p(h) - p) \cdot \mathrm{FPR}_S(h)|\\
& \qquad\qquad \text{(By equation 18 and equation 17)}\\
=& |p - p(h)| \cdot |\mathrm{TPR}_S(h) - \mathrm{FPR}_S(h)| \tag{19}
\end{aligned}
$$

which yields:

$$
d_{\mathrm{TV}}(\mathcal{D}_{Y|S}, \mathcal{D}_Y(h)) - d_{\mathrm{TV}}(\mathcal{D}_{h|S}, \mathcal{D}_h(h)) = |p - p(h)|(1 - |\mathrm{TPR}_S(h) - \mathrm{FPR}_S(h)|)
$$
$$
\text{(By equation 16 and equation 19)}
$$

completing the proof. $\qquad\square$

## A.12 PROOF OF PROPOSITION B.2

*Proof.*

$$
\begin{aligned}
&|p(h_S^*) - p(h_T^*)| \cdot \frac{1}{\mathbb{P}_{\mathcal{D}_S}(Y = +1)} \\
=& \frac{|(1 - \mathrm{Err}_{\mathcal{D}_S}(h_S^*))\mathrm{TPR}_S(h_S^*) - (1 - \mathrm{Err}_{\mathcal{D}_S}(h_T^*))\mathrm{TPR}_S(h_T^*)|}{(1 - \mathrm{Err}_{\mathcal{D}_S}(h_S^*)) \cdot (1 - \mathrm{Err}_{\mathcal{D}_S}(h_T^*))} \\
\leq& \frac{|\mathrm{Err}_{\mathcal{D}_S}(h_S^*) - \mathrm{Err}_{\mathcal{D}_S}(h_T^*)| \cdot |\mathrm{TPR}_S(h_S^*) - \mathrm{TPR}_S(h_T^*)|}{(1 - \mathrm{Err}_{\mathcal{D}_S}(h_S^*)) \cdot (1 - \mathrm{Err}_{\mathcal{D}_S}(h_T^*))}
\end{aligned}
\tag{20}
$$

The inequality above is due to Lemma 7 of Liu & Liu (2015). □

## B LOWER BOUND AND EXAMPLE FOR TARGET SHIFT

### B.1 LOWER BOUND

Now we discuss lower bounds. Denote by $\mathrm{TPR}_S(h)$ and $\mathrm{FPR}_S(h)$ the true positive and false positive rates of $h$ on the source distribution $\mathcal{D}_S$. We prove the following:

**Theorem B.1.** *Under target shift, any model $h$ must incur the following error on either the $\mathcal{D}_S$ or $\mathcal{D}(h)$:*

$$
\begin{aligned}
&\max\{Err_{\mathcal{D}_S}(h), Err_{\mathcal{D}(h)}(h)\} \\
\geq& \frac{|p - p(h)| \cdot (1 - |TPR_S(h) - FPR_S(h)|)}{2}.
\end{aligned}
$$

The proof extends the bound of Theorem 3.3 by further explicating each of $d_{\mathrm{TV}}(\mathcal{D}_{Y|S}, \mathcal{D}_Y(h))$, $d_{\mathrm{TV}}(\mathcal{D}_{h|S}, \text{ and } \mathcal{D}_h(h))$ under the assumption of target shift. Since $|\mathrm{TPR}_S(h) - \mathrm{FPR}_S(h)| < 0$ unless we have a trivial classifier that has either $\mathrm{TPR}_S(h) = 1, \mathrm{FPR}_S(h) = 0$ or $\mathrm{TPR}_S(h) = 0, \mathrm{FPR}_S(h) = 1$, the lower bound is strictly positive. Taking a closer look, the lower bound is determined linearly by how much the label distribution shifts: $p - p(h)$. The difference is further determined by the performance of $h$ on the source distribution through $1 - |\mathrm{TPR}_S(h) - \mathrm{FPR}_S(h)|$. For instance, when $\mathrm{TPR}_S(h) > \mathrm{FPR}_S(h)$, the quality becomes $\mathrm{FNR}_S(h) + \mathrm{FPR}_S(h)$, that is the more error $h$ makes, the larger the lower bound will be.

### B.2 EXAMPLE USING REPLICATOR DYNAMICS

Let us instantiate the discussion using a specific fitness function for the replicator dynamics model (Section 2.1), which is the prediction accuracy of $h$ for class $+1$:

$$
[\textbf{Fitness of } Y = +1] := \mathbb{P}_{\mathcal{D}_S}(h(X) = +1|Y = +1)
\tag{21}
$$

Then we have $\mathbb{E}[\textbf{Fitness of } Y] = \mathrm{Err}_{\mathcal{D}_S}(h)$, and

$$
\frac{p(h)}{\mathbb{P}_{\mathcal{D}_S}(Y = +1)} = \frac{\mathbb{P}_{\mathcal{D}_S}(h(X) = +1|Y = +1)}{\mathrm{Err}_{\mathcal{D}_S}(h)}
$$

Plugging the result back to our Theorem 5.1 we have

**Proposition B.2.** *Under the replicator dynamics model in Eqn. (21), $|p(h_S^*) - p(h_T^*)|$ further bounds as:*

$$
\begin{aligned}
&|p(h_S^*) - p(h_T^*)| \leq \mathbb{P}_{\mathcal{D}_S}(Y = +1) \\
&\cdot \frac{|Err_{\mathcal{D}_S}(h_S^*) - Err_{\mathcal{D}_S}(h_T^*)| \cdot |TPR_S(h_S^*) - TPR_S(h_T^*)|}{Err_{\mathcal{D}_S}(h_S^*) \cdot Err_{\mathcal{D}_S}(h_T^*)}.
\end{aligned}
$$

That is, the difference between $\mathrm{Err}_{\mathcal{D}(h_S^*)}(h_S^*)$ and $\mathrm{Err}_{\mathcal{D}(h_T^*)}(h_T^*)$ is further dependent on the difference between the two classifiers' performances on the source data $\mathcal{D}_S$. This offers an opportunity to evaluate the possible error transferability using the source data only.

## C    MISSING EXPERIMENTAL DETAILS

### C.1    SYNTHETIC EXPERIMENTS USING DAG

**Synthetic experiments using simulated data**    We generate synthetic data sets from structural equation models described on simple causal DAG in Figure 2 for covariate shift and target shift. To generate the induced distribution $\mathcal{D}(h)$, we posit a specific *adaptation function* $\Delta : \mathbb{R}^d \times \mathcal{H} \to \mathbb{R}^d$, so that when an input $x$ encounters classifier $h \in \mathcal{H}$, its induced features are precisely $x' = \Delta(x, h)$. We provide details of the data generation processes and adaptation functions in Appendix C.

We take our training data set $\{x_1, \ldots, x_n\}$ and learn a "base" logistic regression model $h(x) = \sigma(w \cdot x)$[7]. We then consider the hypothesis class $\mathcal{H} := \{h_\tau \mid \tau \in [0,1]\}$, where $h_\tau(x) := 2 \cdot \mathbb{1}[\sigma(w \cdot x) > \tau] - 1$. To compute $h_S^*$, the model that performs best on the source distribution, we simply vary $\tau$ and take the $h_\tau$ with lowest prediction error. Then, we posit a specific adaptation function $\Delta(x, h_\tau)$. Finally, to compute $h_T^*$, we vary $\tau$ from 0 to 1 and find the classifier $h_\tau$ that minimizes the prediction error on its induced data set $\{\Delta(x_1, h_\tau), \ldots, \Delta(x_n, h_\tau)\}$. We report our results in Figure 4.

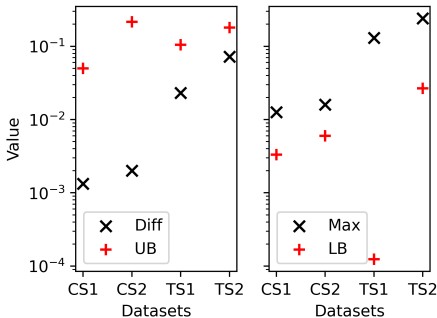

Figure 4: Results for synthetic experiments on simulated and real-world data. $\mathsf{Diff} := \mathrm{Err}_{\mathcal{D}(h_S^*)}(h_S^*) - \mathrm{Err}_{\mathcal{D}(h_T^*)}(h_T^*)$, $\mathsf{Max} := \max\{\mathrm{Err}_{\mathcal{D}_S}(h_T^*), \mathrm{Err}_{\mathcal{D}(h_T^*)}(h_T^*)\}$, $\mathsf{UB} :=$ upper bound specified in Theorem 4.2, and $\mathsf{LB} :=$ lower bound specified in Theorem 4.6.

**Covariate Shift**    We specify the causal DAG for covariate shift setting in the following way:

$$X_1 \sim \mathrm{Unif}(-1, 1)$$
$$X_2 \sim 1.2X_1 + \mathcal{N}(0, \sigma_2^2)$$
$$X_3 \sim -X_1^2 + \mathcal{N}(0, \sigma_3^2)$$
$$Y := 2\mathrm{sign}(X_2 > 0) - 1$$

where $\sigma_2^2$ and $\sigma_3^2$ are parameters of our choices.
*Adaptation function*    We assume the new distribution of feature $X_1'$ will be generated in the following way:

$$X_1' = \Delta(X) = X_1 + c \cdot (h(X) - 1)$$

where $c \in \mathbb{R}^1 > 0$ is the parameter controlling how much the prediction $h(X)$ affect the generating of $X_1'$, namely the magnitude of distribution shift. Intuitively, this adaptation function means that if a feature $x$ is predicted to be positive ($h(x) = +1$), then decision subjects are more likely to adapt to that feature in the induced distribution; Otherwise, decision subjects are more likely to be moving away from $x$ since they know it will lead to a negative prediction.

**Target Shift**    We specify the causal DAG for target shift setting in the following way:

$$(Y + 1)/2 \sim \mathrm{Bernoulli}(\alpha)$$
$$X_1 | Y = y \sim \mathcal{N}_{[0,1]}(\mu_y, \sigma^2)$$
$$X_2 = -0.8X_1 + \mathcal{N}(0, \sigma_2^2)$$
$$X_3 = 0.2Y + \mathcal{N}(0, \sigma_3^2)$$

---

[7]$\sigma(\cdot)$ is the logistic function and $w \in \mathbb{R}^3$ denotes the weights.

where $\mathcal{N}_{[0,1]}$ represents a truncated Gaussian distribution taken value between 0 and 1. $\alpha$, $\mu_y$, $\sigma^2$, $\sigma_2^2$ and $\sigma_3^2$ are parameters of our choices.

*Adaptation function* We assume the new distribution of the qualification $Y'$ will be updated in the following way:

$$\mathbb{P}(Y' = +1 | h(X) = h, Y = y) = c_{hy}, \text{ where } \{h, y\} \in \{-1, +1\}$$

where $0 \leq c_{hy} \in \mathbb{R}^1 \leq 1$ represents the likelihood for a person with original qualification $Y = y$ and get predicted as $h(X) = h$ to be qualified in the next step ($Y' = +1$).

**Discussion of the Results** For all four datasets, we do observe positive gaps $\text{Err}_{D(h_S^*)}(h_S^*) - \text{Err}_{D(h_T^*)}(h_T^*)$, indicating the suboptimality of training on $\mathcal{D}_S$. The gaps are well bounded by the theoretical results. For lower bound, the empirical observation and the theoretical bounds are roughly within the same magnitude except for one target shift dataset, indicating the effectiveness of our theoretical result. For upper bound, for target shift, the empirical observations are well within the same magnitude of the theoretical bounds while the results for the covariate shift are relatively loose.

## C.2 SYNTHETIC EXPERIMENTS USING REAL-WORLD DATA

On the preprocessed FICO credit score data set (Board of Governors of the Federal Reserve System (US), 2007; Hardt et al., 2016b), we convert the cumulative distribution function (CDF) of TransRisk score among demographic groups (denoted as $A$, including Black, Asian, Hispanic, and White) into group-dependent densities of the credit score. We then generate a balanced sample where each group has equal representation, with credit scores (denoted as $Q$) initialized by sampling from the corresponding group-dependent density. The value of attributes for each data point is then updated under a specified dynamics (detailed in Appendix C.2.1) to model the real-world scenario of repeated resource allocation (with decision denoted as $D$).

### C.2.1 PARAMETERS FOR DYNAMICS

Since we are considering the dynamic setting, we further specify the data generating process in the following way (from time step $T = t$ to $T = t + 1$):

$$
\begin{aligned}
X_{t,1} &\sim 1.5 Q_t + U[-\epsilon_1, \epsilon_1] \\
X_{t,2} &\sim 0.8 A_t + U[-\epsilon_2, \epsilon_2] \\
X_{t,3} &\sim A_t + \mathcal{N}(0, \sigma^2) \\
Y_t &\sim \text{Bernoulli}(q_t) \text{ for a given value of } Q_t = q_t \\
D_t &= f_t(A_t, X_{t,1}, X_{t,2}, X_{t,3}) \\
Q_{t+1} &= \{Q_t \cdot [1 + \alpha_D(D_t) + \alpha_Y(Y_t)]\}_{(0,1]} \\
A_{t+1} &= A_t \text{ (fixed population)}
\end{aligned}
$$

where $\{\cdot\}_{(0,1]}$ represents truncated value between the interval $(0, 1]$, $f_t(\cdot)$ represents the decision policy from input features, and $\epsilon_1, \epsilon_2, \sigma$ are parameters of choices. In our experiments, we set $\epsilon_1 = \epsilon_2 = \sigma = 0.1$.

Within the same time step, i.e., for variables that share the subscript $t$, $Q_t$ and $A_t$ are root causes for all other variables ($X_{t,1}, X_{t,2}, X_{t,3}, D_t, Y_t$). At each time step $T = t$, the institution first estimates the credit score $Q_t$ (which is not directly visible to the institution, but is reflected in the visible outcome label $Y_t$) based on ($A_t, X_{t,1}, X_{t,2}, X_{t,3}$), then produces the binary decision $D_t$ according to the optimal threshold (in terms of the accuracy).

For different time steps, e.g., from $T = t$ to $T = t + 1$, the new distribution at $T = t + 1$ is induced by the deployment of the decision policy $D_t$. Such impact is modeled by a multiplicative update in $Q_{t+1}$ from $Q_t$ with parameters (or functions) $\alpha_D(\cdot)$ and $\alpha_Y(\cdot)$ that depend on $D_t$ and $Y_t$, respectively. In our experiments, we set $\alpha_D = 0.01$ and $\alpha_Y = 0.005$ to capture the scenario where one-step influence of the decision on the credit score is stronger than that for ground truth label.

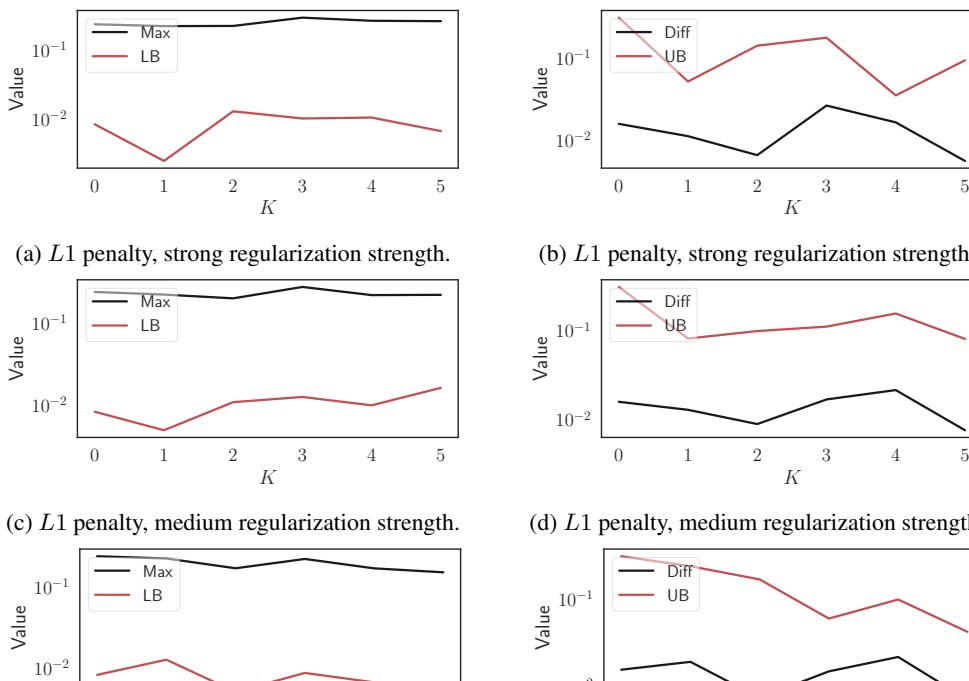

Figure 5: Results of applying $L1$ penalty with different strength when constructing $h_S^*$. The left column consisting of panels (a), (c), and (e) compares $\mathsf{Max} := \max\{\mathrm{Err}_{\mathcal{D}_S}(h_T^*), \mathrm{Err}_{\mathcal{D}(h_T^*)}(h_T^*)\}$ and $\mathsf{LB} :=$ lower bound specified in Theorem 4.6. The right column consisting of panels (b), (d), and (f) compares $\mathsf{Diff} := \mathrm{Err}_{\mathcal{D}(h_S^*)}(h_S^*) - \mathrm{Err}_{\mathcal{D}(h_T^*)}(h_T^*)$ and $\mathsf{UB} :=$ upper bound specified in Theorem 4.2. For each time step $K = k$, we compute and deploy the source optimal classifier $h_S^*$ and update the credit score for each individual according to the received decision as the new reality for time step $K = k + 1$.

## C.2.2 ADDITIONAL EXPERIMENTAL RESULTS

In this section, we present additional experimental results on the real-world FICO credit score data set. With the initialization of the distribution of credit score $Q$ and the specified dynamics, we present results comparing the influence of vanilla regularization terms in decision-making (when estimating the credit score $Q$) on the calculation of bounds for induced risks.[8] In particular, we consider $L1$ norm (Figure 5) and $L2$ norm (Figure 6) regularization terms when optimizing decision-making policies on the source domain. As we can see from the results, applying vanilla regularization terms (e.g., $L1$ norm and $L2$ norm) on source domain without specific considerations of the inducing-risk mechanism does not provide significant performance improvement in terms of smaller induced risk. For example, there is no significant decrease of the term $\mathsf{Diff}$ as the regularization strength increases, for both $L1$ norm (Figure 5) and $L2$ norm (Figure 6) regularization terms.

---

[8]The regularization that involves induced risk considerations will be discussed in Appendix F.

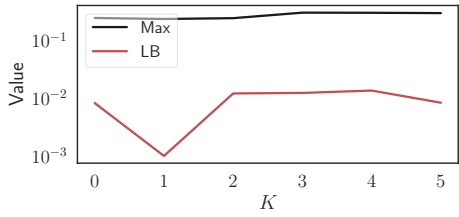

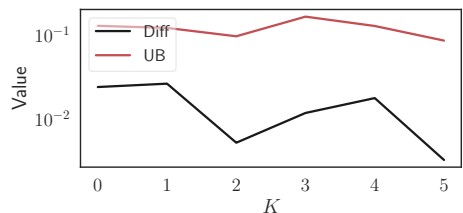

(a) $L2$ penalty, strong regularization strength.

(b) $L2$ penalty, strong regularization strength.

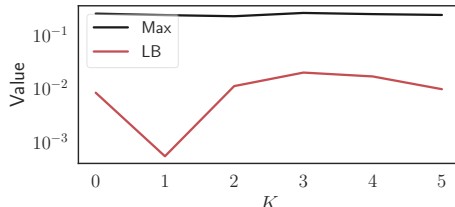

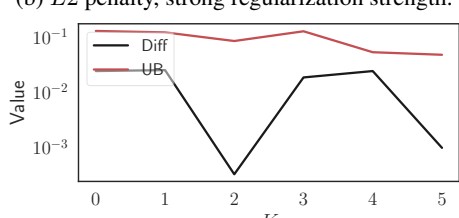

(c) $L2$ penalty, medium regularization strength.

(d) $L2$ penalty, medium regularization strength.

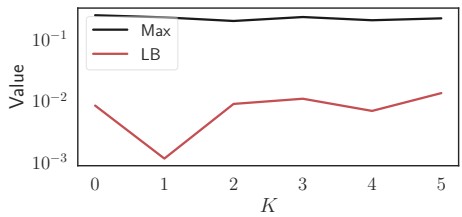

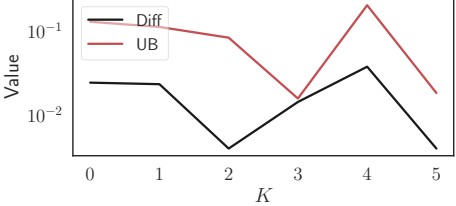

(e) $L2$ penalty, weak regularization strength.

(f) $L2$ penalty, weak regularization strength.

Figure 6: Results of applying $L2$ penalty with different strength when constructing $h_S^*$. The left column consisting of panels (a), (c), and (e) compares Max $:=$ $\max\{\mathrm{Err}_{\mathcal{D}_S}(h_T^*), \mathrm{Err}_{\mathcal{D}(h_T^*)}(h_T^*)\}$ and LB $:=$ lower bound specified in Theorem 4.6. The right column consisting of panels (b), (d), and (f) compares Diff $:= \mathrm{Err}_{\mathcal{D}(h_S^*)}(h_S^*) - \mathrm{Err}_{\mathcal{D}(h_T^*)}(h_T^*)$ and UB $:=$ upper bound specified in Theorem 4.2. For each time step $K = k$, we compute and deploy the source optimal classifier $h_S^*$ and update the credit score for each individual according to the received decision as the new reality for time step $K = k + 1$.

# D CHALLENGES IN MINIMIZING INDUCED RISK

## D.1 COMPUTATIONAL CHALLENGES

The literature of domain adaptation has provided us solutions to minimize the risk on the target distribution via a nicely developed set of results Sugiyama et al. (2008; 2007); Shimodaira (2000). This allows us to extend the solutions to minimize the induced risk too. Nonetheless we will highlight additional computational challenges.

We focus on the covariate shift setting. The scenario for target shift is similar. For covariate shift, recall that earlier we derived the following fact:

$$\mathbb{E}_{\mathcal{D}(h)}[\ell(h; X, Y)] = \mathbb{E}_{\mathcal{D}}[\omega_x(h) \cdot \ell(h; x, y)]$$

This formula informs us that a promising solution that uses $\omega_x(h)$ to perform reweighted ERM. Of course, the primary challenge that stands in the way is how do we know $\omega_x(h)$. There are different methods proposed in the literature to estimate $\omega_x(h)$ when one has access to $\mathcal{D}(h)$ Zhang et al. (2013b); Long et al. (2016); Gong et al. (2016). How any of the specific techniques work in our induced domain adaptation setting will be left for a more thorough future study. In this section, we focus on explaining the computational challenges even when such knowledge of $\omega_x(h)$ can be obtained for each model $h$ being considered during training.

Though $\omega_x(h), \ell(h; x, y)$ might both be convex with respect to (the output of) the classifier $h$, their product is not necessarily convex. Consider the following example:

*Example* 1 ($\omega_x(h) \cdot \ell(h; x, y)$ is generally non-convex). Let $\mathcal{X} = (0, 1]$. Let the true label of each $x \in \mathcal{X}$ be $y(x) = \mathbb{1}\left(x \geq \frac{1}{2}\right)$. Let $\ell(h; x, y) = \frac{1}{2}(h(x) - y)^2$, and let $h(x) = x$ (simple linear model). Notice that $\ell$ is convex in $h$. Let $\mathcal{D}$ be the uniform distribution, whose density function is $f_{\mathcal{D}} = \begin{cases} 1, & 0 < x \leq 1 \\ 0, & \text{otherwise} \end{cases}$. Notice that if the training data is drawn from $\mathcal{D}$, then $h$ is the linear classifier that minimizes the expected loss. Suppose that, since $h$ rewards large values of $x$, it induces decision subjects to shift towards higher feature values. In particular, let $\mathcal{D}(h)$ have density function

$$f_{\mathcal{D}(h)} = \begin{cases} 2x, & 0 < x \leq 1 \\ 0, & \text{otherwise} \end{cases}$$

Then for all $x \in \mathcal{X}$, $\omega_x(h) = \frac{f_{\mathcal{D}(h)}(x)}{f_{\mathcal{D}}(x)} = 2x$. Notice that $\omega_x(h) = 2x$ is convex in $h(x) = x$. Then

$$\omega_x(h) \cdot \ell(h; x, y) = 2x \cdot \frac{1}{2}(h(x) - y)^2$$

$$= x(x - y)^2 = \begin{cases} x^3, & 0 < x < \frac{1}{2} \\ x(x - 1)^2, & \frac{1}{2} \leq x \leq 1 \end{cases}$$

which is clearly non-convex.

Nonetheless, we provide sufficient conditions under which $\omega_x(h) \cdot \ell(h; x, y)$ is in fact convex:

**Proposition D.1.** *Suppose $\omega_x(h)$ and $\ell(h; x, y)$ are both convex in $h$, and $\omega_x(h)$ and $\ell(h; x, y)$ satisfy $\forall h, h', x, y:$ $(\omega_x(h) - \omega_x(h')) \cdot (\ell(h; x, y) - \ell(h'; x, y)) \geq 0$. Then $\omega_x(h) \cdot \ell(h; x, y)$ is convex.*

*Proof.* Let us use the shorthand $\omega(h) := \omega_x(h)$ and $\ell(h) := \ell(h; x, y)$. To show that $\omega(h) \cdot \ell(h)$ is convex, it suffices to show that for any $\alpha \in [0, 1]$ and any two hypotheses $h, h'$ we have

$$\omega(\alpha \cdot h + (1 - \alpha) \cdot h') \cdot \ell(\alpha \cdot h + (1 - \alpha) \cdot h') \leq \alpha \cdot \omega(h) \cdot \ell(h) + (1 - \alpha) \cdot \omega(h') \cdot \ell(h')$$

By the convexity of $\omega$,

$$\omega(\alpha \cdot h + (1 - \alpha) \cdot h') \leq \alpha \cdot \omega(h) + (1 - \alpha) \cdot \omega(h')$$

and by the convexity of $\ell$,

$$\ell(\alpha \cdot h + (1 - \alpha) \cdot h') \leq \alpha \cdot \ell(h) + (1 - \alpha) \cdot \ell(h')$$

Therefore it suffices to show that

$$[\alpha \cdot \omega(h) + (1 - \alpha) \cdot \omega(h')] \cdot [\alpha \cdot \ell(h) + (1 - \alpha) \cdot \ell(h')] - \alpha \cdot \omega(h) \cdot \ell(h) + (1 - \alpha) \cdot \omega(h') \cdot \ell(h') \leq 0$$

$$\Leftrightarrow \alpha(\alpha - 1) \cdot \omega(h)\ell(h) - \alpha(\alpha - 1) \cdot [\omega(h)\ell(h') + \omega(h')\ell(h)] + \alpha(\alpha - 1) \cdot \omega(h')\ell(h') \leq 0$$

$$\Leftrightarrow \alpha(\alpha - 1) \cdot [\omega(h) - \omega(h')] \cdot [\ell(h) - \ell(h')] \leq 0$$

$$\Leftrightarrow [\omega(h) - \omega(h')] \cdot [\ell(h) - \ell(h')] \geq 0$$

By the assumed condition, the left-hand side is indeed non-negative, which proves the claim. $\square$

This condition is intuitive when each $x$ belongs to a rational agent who responds to a classifier $h$ to maximize her chance of being classified as $+1$: For $y = +1$, the higher loss point corresponds to the ones that are close to decision boundary, therefore, more $-1$ negative label points might shift to it, resulting to a larger $\omega_x(h)$. For $y = -1$, the higher loss point corresponds to the ones that are likely mis-classified as +1, which "attracts" instances to deviate to.

## D.2 CHALLENGES DUE TO THE LACK OF ACCESS TO DATA

We discuss the challenges in performing induced domain adaptation. In the standard domain adaptation settings, one often assumes the access to a sample set of $X$, which already poses challenges

when there is no access to label $Y$ after the adaptation. Nonetheless, the literature has observed a fruitful development of solutions Sugiyama et al. (2008); Zhang et al. (2013b); Gong et al. (2016).

One might think the above idea can be applied to our IDA setting rather straightforwardly by assuming observing samples from $\mathcal{D}(h)$, the induced distribution under each model $h$ during the training. However, we often do not know precisely how the distribution would shift under a model $h$ until we deploy it. This is particularly true when the distribution shifts are caused by human responding to a model. Therefore, the ability to "predict" accurately how samples "react" to $h$ plays a very important role Ustun et al. (2019). Indeed, the strategic classification literature enables this capability by assuming full rational human agents. For a more general setting, building robust domain adaptation tools that are resistant to the above "prediction error" is also going to be a crucial criterion.

# E   DISCUSSIONS ON PERFORMING DIRECT INDUCED RISK MINIMIZATION

In this section, we provide discussions on how to directly perform induced risk minimization for our induced domain adaptation setting. We first provide a gradient descent based method for a particular label shift setting where the underlying dynamic is replicator dynamic described in Section 5.3. Then we propose a solution for a more general induced domain adaptation setting where we do not make any particular assumptions on the undelying distribution shift model.

## E.1   GRADIENT DESCENT BASED METHOD

Here we provide a toy example of performing direct induced risk minimization under the assumption of label shift with underlying dynamics as the replicator dynamics described in Section 5.3.

**Setting**   Consider a simple setting in which each decision subject is associated with a 1-dimensional continuous feature $x \in \mathbb{R}$ and a binary true qualification $y \in \{-1, +1\}$. We assume label shift setting, and the underlying population dynamic evolves the replicator dynamic setting described in Section 5.3. We consider a simple threshold classifier, where $\hat{Y} = h(x) = 1[X \geq \theta]$, meaning that the classifier is completely characterized by the threshold parameter $\theta$. Below we will use $\hat{Y}$ and $h(X)$ interchangeably to represent the classification outcome. Recall that the replicator dynamics is specified as follows:

$$\frac{\mathbb{P}_{\mathcal{D}(h)}(Y = y)}{\mathbb{P}_{\mathcal{D}_S}(Y = y)} = \frac{\textbf{Fitness}(Y = y)}{\mathbb{E}_{\mathcal{D}_S}[\textbf{Fitness}(Y)]} \tag{22}$$

where $\mathbb{E}_{\mathcal{D}_S}[\textbf{Fitness}(Y)] = \textbf{Fitness}(Y = y)\mathbb{P}_{\mathcal{D}_S}(Y = y) + \textbf{Fitness}(Y = -y)(1 - \mathbb{P}_{\mathcal{D}_S}(Y = y))$. $\textbf{Fitness}(Y = y)$ is the fitness of strategy $Y = y$, which is further defined in terms of the expected utility $U_{y,\hat{y}}$ of each qualification-classification outcome pair $(y, \hat{y})$:

$$\textbf{Fitness}(Y = y) := \sum_{\hat{y}} \mathbb{P}[\hat{Y} = \hat{y} | Y = y] \cdot U_{y,\hat{y}}$$

where $U_{y,\hat{y}}$ is the utility (or reward) for each qualification-classification outcome combination. $\mathbb{P}(X | Y = y)$ is sampled according to a Gaussian distribution, and will be unchanged since we consider a label shift setting.

We initialize the distributions we specify the initial qualification rate $\mathbb{P}_{\mathcal{D}_S}(Y = +1)$. To test different settings, we vary the specification of the utility matrix $U_{y,\hat{y}}$ and generate different dynamics.

**Formulate the induced risk as a function of $h$**   To minimize the induced risk, we first formulate the induced risk as a function of the classifier $h$'s parameter $\theta$ taking into account of the underlying dynamic, and then perform gradient descent to solve for locally optimal classifier $h_T^*$.

Recall from Section 5, under label shift, we can rewrite the induced risk as the following form:

$$\mathbb{E}_{\mathcal{D}(h)}[\ell(h; X, Y)] = p(h) \cdot \mathbb{E}_{\mathcal{D}_S}[\ell(h; X, Y)|Y = +1] + (1 - p(h)) \cdot \mathbb{E}_{\mathcal{D}_S}[\ell(h; X, Y)|Y = -1]$$

where $p(h) = \mathbb{P}_{\mathcal{D}(h)}(Y = +1)$.

Since $\mathbb{E}_{\mathcal{D}_S}[\ell(h; X, Y)|Y = +1]$ and $\mathbb{E}_{\mathcal{D}_S}[\ell(h; X, Y)|Y = -1]$ are already functions of both $h$ and $\mathcal{D}_S$, it suffices to show that the accuracy on $\mathcal{D}(h)$, $p(h) = \mathbb{P}_{\mathcal{D}(h)}(Y = +1)$, can also be expressed as a function of $\theta$ and $\mathcal{D}_S$.

To see this, recall that for a threshold classifier $\hat{Y} = 1[X > \theta]$, it means that the prediction accuracy can be written as a function of the threshold $\theta$ and target distribution $\mathcal{D}(h)$:

$$
\begin{aligned}
&\mathbb{P}_{\mathcal{D}(h)}(Y = +1) \\
&= \mathbb{P}_{\mathcal{D}(h)}(\hat{Y} = +1, Y = +1) + \mathbb{P}_{\mathcal{D}(h)}(\hat{Y} = -1, Y = -1) \\
&= \mathbb{P}_{\mathcal{D}(h)}(X \geq \theta, Y = +1) + \mathbb{P}_{\mathcal{D}(h)}(X \leq \theta, Y = -1) \\
&= \int_{\theta}^{\infty} \mathbb{P}_{\mathcal{D}(h)}(Y = +1) \underbrace{\mathbb{P}(X = x | Y = 1)}_{\text{unchanged because of label shift}} dx \\
&\quad + \int_{-\infty}^{\theta} \mathbb{P}_{\mathcal{D}(h)}(Y = -1) \underbrace{\mathbb{P}(X = x | Y = -1)}_{\text{unchanged because of label shift}} dx
\end{aligned}
\tag{23}
$$

where $\mathbb{P}(X | Y = y)$ remains unchanged over time, and $\mathbb{P}_{\mathcal{D}(h)}(Y = y)$ evolves over time according to Equation (22), namely

$$
\begin{aligned}
&\mathbb{P}_{\mathcal{D}(h)}(Y = y) \\
&= \mathbb{P}_{\mathcal{D}_S}(Y = y) \times \frac{\mathbf{Fitness}_g(Y = y)}{\mathbb{E}_{\mathcal{D}_S}[\mathbf{Fitness}_g(Y)]} \\
&= \mathbb{P}_{\mathcal{D}_S}(Y = y) \times \frac{\sum_{\hat{y}} \mathbb{P}_{\mathcal{D}_S}[\hat{Y} = \hat{y} | Y = y, G = g] \cdot U_{\hat{y},y}}{\sum_y (\sum_{\hat{y}} \mathbb{P}_{\mathcal{D}_S}[\hat{Y} = \hat{y} | Y = y, G = g] \cdot U_{\hat{y},y}) \mathbb{P}_{\mathcal{D}_S}[Y = y]}
\end{aligned}
\tag{24}
$$

Notice that $\hat{Y}$ is only a function of $\theta$, and $U_{y,\hat{y}}$ are fixed quantities, the above derivation indicates that we can express $\mathbb{P}_{\mathcal{D}(h)}(Y = y)$ as a function of $\theta$ and $\mathcal{D}_S$. Plugging it back to Equation (23), we can see that the accuracy can also be expressed as a function of the classifier's parameter $\theta$, indicating that the induced risk can be expressed as a function of $\theta$. Thus we can use gradient descent using automatic differentiation w.r.t $\theta$ to find a optimal classifier $h_T^*$ that minimize the induced risk.

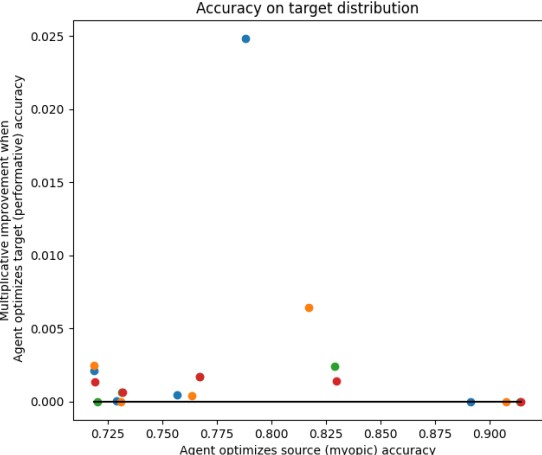

Figure 7: Experimental results of directly optimizing for the induced risk under the assumption of replicator dynamic. The X-axis denotes the prediction accuracy of $\mathrm{Err}_{D(h_S^*)}(h_S^*)$, where $h_S^*$ is the source optimal classifier under each settings. The Y-axis is the percent of performance improvement using the classifier that optimize for $h_T^* = \arg\min \mathrm{Err}_{D(h)}(h)$, which the decision maker considers the underlying response dynamics (according to replicator dynamics in Equation (22)) of the decision subjects. Different color represents different utility function, which is reflected by the specifications of values in $U_{y,\hat{y}}$; within each color, different dots represent different initial qualification rate.

**Experimental Results** Figure 7 shows the experimental results for this toy example. We can see that for each setting, compared to the baseline classifier $h_S^*$, the proposed gradient based optimization

procedure returns us a classifier that achieves a better prediction accuracy (thus lower induced risk) compared to the accuracy of the source optimal classifier.

## E.2 GENERAL SETTING: INDUCED RISK MINIMIZATION WITH BANDIT FEEDBACK

In general, finding the optimal classifier that achieves the optimal induced risk $h_T^*$ is a hard problem due to the interactive nature of the problem (see, e.g. the literature of performative prediction Perdomo et al. (2020) for more detailed discussions). Without making any assumptions on the mapping between $h$ and $\mathcal{D}(h)$, one can only potentially rely on the *bandit feedbacks* from the decision subjects to estimate the influence of $h$ on $\mathcal{D}(h)$: when the induced risk is a convex function of the classifier $h$'s parameter $\theta$, one possible approach is to use the standard techniques from bandit optimization (Flaxman et al., 2004) to iteratively find induced optimal classifier $h_T^*$. The basic idea is: at each step $t = 1, \cdots, T$, the decision maker deploy a classifier $h_t$, then observe data points sampled from $\mathcal{D}(h_t)$ and their losses, and use them to construct an approximate gradient for the induced risk as a function of the model parameter $\theta_t$. When the induced risk is a convex function in the model parameter $\theta$, the above approach guarantees to converge to $h_T^*$, and have sublinear regret in the total number of steps $T$.

The detailed description of the algorithm for finding $h_T^*$ is as follows:

---
**Algorithm 1:** One-point bandit gradient descent for performative prediction

---
**Result:** return $\theta_T$ after $T$ rounds
$\theta_1 \leftarrow 0$
**foreach** *time step* $t \leftarrow 1, \ldots, T$ **do**
    Sample a unit vector $u_t \sim \text{Unif}(\mathbf{S})$
    $\theta_t^+ \leftarrow \theta_t + \delta u_t$
    Observe data points $z_1, \ldots, z_{n_t} \sim \mathcal{D}(\theta_t^+)$
    $\widetilde{\mathsf{IR}}(\theta_t^+) \leftarrow \frac{1}{n_t} \sum_{i=1}^{n_t} \ell(z_i; \theta_t^+)$
    $\tilde{g}_t(\theta_t) \leftarrow \frac{d}{\delta} \widetilde{\mathsf{IR}}(\theta_t^+) \cdot u_t$            $\triangleright \tilde{g}_t(\theta_t)$ *is an approximation of* $\nabla_\theta \widehat{\mathsf{IR}}(\theta_t)$
    $\theta_{t+1} \leftarrow \Pi_{(1-\delta)\Theta}(\theta_t - \eta \tilde{g}_t(\theta_t))$        $\triangleright$ *Take gradient step; project onto*
    $(1 - \delta)\Theta := \{(1 - \delta)\theta \mid \theta \in \Theta\}$
**end**

---

## F REGULARIZED TRAINING

In this section, we discuss the possibility that indeed minimizing regularized risk will lead to a tighter upper bound. Consider the target shift setting. Recall that $p(h) := \mathbb{P}_{\mathcal{D}(h)}(Y = +1)$ and we have for any proper loss function $\ell$:

$$\mathbb{E}_{\mathcal{D}(h)}[\ell(h; X, Y)] = p(h) \cdot \mathbb{E}_{\mathcal{D}_S}[\ell(h; X, Y)|Y = +1] + (1 - p(h)) \cdot \mathbb{E}_{\mathcal{D}_S}[\ell(h; X, Y)|Y = -1]$$

Suppose $p < p(h_T^*)$, now we claim that minimizing the following regularized/penalized risk leads to a smaller upper bound.

$$\mathbb{E}_{\mathcal{D}_S}[\ell(h; X, Y)] + \alpha \cdot \mathbb{E}_{\mathcal{D}_{\text{uniform}}}||\frac{h(X) + 1}{2}||$$

where in above $\mathcal{D}_{\text{uniform}}$ is a distribution with uniform prior for $Y$.

We impose the following assumption:

- The number of predicted $+1$ for examples with $Y = +1$ and for examples with $Y = -1$ are monotonic with respect to $\alpha$.

Consider the easier setting with $\ell = 0$-1 loss. Then

$$\mathbb{E}_{\mathcal{D}_{\text{uniform}}}||h(X)|| = 0.5 \cdot (\mathbb{P}_{X|Y=+1}[h(X) = +1] + \mathbb{P}_{X|Y=-1}[h(X) = +1]) - 0.5$$
$$= 0.5 \cdot (\mathbb{E}_{X|Y=+1}[\ell(h(X), +1)] - \mathbb{E}_{X|Y=-1}[\ell(h(X), -1)])$$

The above regularized risk minimization problem is equivalent to

$$(p + 0.5 \cdot \alpha) \cdot \mathbb{E}_{X|Y=+1}[\ell(h(X), +1)] + (p - 0.5 \cdot \alpha) \cdot \mathbb{E}_{X|Y=-1}[\ell(h(X), -1]$$

Recall the upper bound in Theorem 5.1:

$$\text{Err}_{\mathcal{D}(h_S^*)}(h_S^*) - \text{Err}_{\mathcal{D}(h_T^*)}(h_T^*) \leq \underbrace{|p(h_S^*) - p(h_T^*)|}_{\text{Term 1}}$$
$$+ (1 + p) \cdot \underbrace{(d_{\text{TV}}(\mathcal{D}_+(h_S^*), \mathcal{D}_+(h_T^*)) + d_{\text{TV}}(\mathcal{D}_-(h_S^*), \mathcal{D}_-(h_T^*)))}_{\text{Term 2}}.$$

With a properly specified $\alpha > 0$, this leads to a distribution with a smaller gap of $|p(\tilde{h}_S) - p(h_T^*)|$, where $\tilde{h}_S$ denotes the optimal classifier of the penalized risk minimization - this leads to a smaller Term 1 in the bound of Theorem 5.1. Furthermore, the induced risk minimization problem will correspond to an $\alpha$ s.t. $\alpha^* = \frac{p(h_T^*) - p}{0.5}$, and the original $h_S^*$ corresponds to a distribution of $\alpha = 0$. Using the monotonicity assumption, we will establish that the second term in Theorem 5.1 will also smaller when we tune a proper $\alpha$.

## G    DISCUSSION ON THE TIGHTNESS OF OUR THEORETICAL BOUNDS

**General Bounds in Section 3**    For the general bounds reported in Section 3, it is not trivial to fully quantify the tightness without further quantifying the specific quantities of the terms, e.g. the H divergence of the source and the induced distribution, and the average error a classifier have to incur for both distribution. This part of our results adapted from the classical literature in learning from multiple domains Ben-David et al. (2010). The tightness of using $\mathcal{H}$-divergence and other terms seem to be partially validated therein.

**Bounds in Section 4 and Section 5**    For more specific bounds provided in Section 4 (for covariate shift) and Section 5 (target shift), however, it is relatively easier to argue about the tightness: the proofs there are more transparent and are easier to back out the conditions where the inequalities are relaxed. For example, in Theorem 5.1, the inequalities of our bound are introduced primarily in the following two places: 1) one is using the optimiality of $h_S^*$ on the source distribution. 2) the other is bounding the statistical difference in $h_S^*$ and $h_T^*$'s predictions on the positive and negative examples. Both are saying that if the differences in the two classifiers' predictions are bounded in a range, then the result in Theorem 5.1 is relatively tight.

