# OpenReview forum: "Model Transferability with Responsive Decision Subjects "
_ICLR.cc/2023/Conference — Submitted to ICLR 2023_

### Official Review · Reviewer_tvWE · 2022-10-24

**Confidence:** 3
**Correctness:** 3
**Technical Novelty And Significance:** 3
**Empirical Novelty And Significance:** 2
**Recommendation:** 5

**Clarity, Quality, Novelty And Reproducibility:**

Clarity: decent clarity around the exposition and proofs, some confusion around replicator dynamics and experiments
Quality: I think the work is solid, however I'm not sure that the assumptions relied on by the proofs are all appropriate
Novelty: I think this is novel because it uses lighter assumptions (mostly) than previous strategic classification work
Reproducibility: I have a little confusion around the experiments at the moment, with a bit more information this could be reproducible

**Strength And Weaknesses:**

Strengths:
- These are useful results and I think more general than what has previously been shown in strategic classification
- paper is mostly pretty clear and exposition is good

Feedback:
-Sec 4 and 5: it isn't clear to me why Covariate/Target shift would be useful assumptions for the strategic classification setting. One would imagine that the induced distribution will have a much lower P(Y = 1 | X) for X which are right over the decision boundary, for instance; or that P(X | Y = 1) would cluster more closely near the decision boundary in the induced distribution. I'm happy to be convinced otherwise, but it seems to me that by the definition of strategic classification, these results aren't so useful.
-Sec 4.2: it would be good to explain why these might be natural assumptions: I think I understand it has something to do with people choosing to move their inputs towards the desirable outcome Y=1, but not totally sure
Sec 4.3, Setup 3/4: I think there needs to be a little more definition around how the adapted feature is generated. For instance, if \tau = 0.5, x = 0.4, and B = 0.2, then the probability of a successful update is 0.5, but it is drawn from the distribution U(0.5, 0,3), which doesn't obviously make sense to me.
-Prop 4.7: I don't quite understand why as Err_D_S(h*_T) goes up in this bound, the gap increases - it would be nice to get some intuition on this relationship
Replicator Dynamics: It would be good to have a better description of this in the body of the paper, I'm not so clear on exactly how this works
Experiments: I don't totally understand the data generation process, even after looking at the supplement. It would be good to have a clearer explanation of the exact strategic modification model here (e.g. what is \epsilon or \alpha, as defined in the supplement)

Notes:
Eq (3) - would be good to define the optimal model here: it's not clear at this point in the paper if h*_T is the model with the optimal induced risk, or optimal source risk (you explain later, but should explain here)
Sec 4.2 - should define X_+(h) and X_-(h)
Sec 5: a little bit confusing to me to overload w here as the induced positive rate, when it was the weighting coefficient earlier
Thm 5.2: what is p here?




**Summary Of The Paper:**

In this paper, the authors explore what guarantees we can identify on the performance of classifiers in response to classification subjects which act strategically - that is, attempt to update their inputs so as to achieve a better output. They call this loss the "induced risk": the risk the model incurs on the input distribution which its deployment induces. Their results are mostly of two forms: upper bounds on the induced risk above the optimal induced risk, and lower bounds on the minimum of the source risk and the induced risk. The machinery used for their proofs is mostly domain-adaptation-inspired, where the source distribution is the original data, and the target distribution is the strategically modified data.

**Summary Of The Review:**

Overall, this paper provides some useful results in the strategic classification space. I have some doubts about the applicability of the covariate shift/label shift assumptions to this setting, and therefore not sure how useful those results are. Additionally, I have some confusion around other assumptions made around the data generative process with replicator dynamics and the experiments. Due to some good results and exposition alongside these concerns, I'm recommending a Weak Reject for now.

---

> ### Author Response · Authors · 2022-11-13
> **Response to Reviewer tvWE (1/2)**
>
> Thank you for the feedback. Below we address your comments and questions:
>
> > Sec 4 and 5: it isn't clear to me why Covariate/Target shift would be useful assumptions for the strategic classification setting. One would imagine that the induced distribution will have a much lower P(Y = 1 | X) for X which are right over the decision boundary, for instance; or that P(X | Y = 1) would cluster more closely near the decision boundary in the induced distribution..
>
> This is a great question; please allow us to clarify further. Under the assumption made in the classific strategic classification setting, changing the feature vector $X$ doesn’t change the underlying true qualification $Y$,  you are totally correct that the induced distribution will have a much lower $P(Y = 1 | X)$ for $X$ which are right over the decision boundary.
> However, in this paper, we consider a slightly different setting where we assume that feature changes $X$ could potentially lead to changes in the true qualification $Y$ and that the mapping between $Y$ and $X$ remains the same before and after the adaptation. This is a common assumption made in a line of recent papers that aims to incentivize improvement behavior from the agents. We have clarified the difference between our setting and the classific strategic classification setting in our revised paper.
>
> >  Sec 4.2: it would be good to explain why these might be natural assumptions: I think I understand it has something to do with people choosing to move their inputs towards the desirable outcome Y=1, but not totally sure
>
> This is a great suggestion. Your intuition is correct, that the three assumptions we make in section 4.2 aim to capture the fact that people are more inclined to move to those features that will have higher chances of being classified as $ +1$, and the corresponding $w_h(X)$ for these features, the shifting ratio, will be higher.
>
> For instance, in the standard strategic classification setting, users are modeled to be strategic about moving to features with a higher chance of being classified as $+1$ – this is the intuition for Assumption 4.4.  In later works (see, e.g. the references mentioned below), the game between the agent and the machine learning models has been designed to induce agents to take actions/changes to move from feature $X$ to $X’$ that can also improve the true qualification $Y$ –  this is the intuition for Assumption 4.3. Assumption 4.5 further requires the above for Xs that are predicted differently.
>
> References:
>
> [Chen et al, 2020] Chen, Yatong, Jialu Wang, and Yang Liu. "Linear Classifiers that Encourage Constructive Adaptation." arXiv preprint arXiv:2011.00355 (2020).
>
> [Haghtalab et al, 2020] Haghtalab, Nika, Nicole Immorlica, Brendan Lucier, and Jack Z. Wang. "Maximizing welfare with incentive-aware evaluation mechanisms." arXiv preprint arXiv:2011.01956 (2020).
>
> [Shavit et al, 2020] Shavit, Yonadav, Benjamin Edelman, and Brian Axelrod. "Causal strategic linear regression." In International Conference on Machine Learning, pp. 8676-8686. PMLR, 2020.
>
>
> > Sec 4.3, Setup 3/4: I think there needs to be a little more definition around how the adapted feature is generated. For instance, if \tau = 0.5, x = 0.4, and B = 0.2, then the probability of a successful update is 0.5, but it is drawn from the distribution U(0.5, 0,3), which doesn't obviously make sense to me.
>
> Great catch. We believe that this is a typo; the new feature distribution should be $U(\tau_h, \tau_h + |B - x|)$. Thank you for pointing it out. Reworking the example,  $\tau = 0.5, x = 0.4, B = 0.2$, then the probability of a successful update is 0.5, but it is drawn from the distribution $\tau_h + |B - x| = 0.5 + |0.2 - 0.4| = 0.7$, thus the new data point $X’ \sim U(0.5, 0,7)$. We have fixed it in our revised paper.

---

> > ### Author Response · Authors · 2022-11-13
> > **Response to Reviewer tvWE (2/2)**
> >
> > > 4.7: I don't quite understand why as $Err\_{\mathcal{D}_S}(h^*_T)$ goes up in this bound, the gap increases - it would be nice to get some intuition on this relationship Replicator Dynamics: It would be good to have a better description of this in the body of the paper, I'm not so clear on exactly how this works
> >
> > In the derivation we have, a covariance term between $h^*_S$ and the shifted distribution is introduced to make the connection between the induced risk of $h^*_S$ and $h^*_T$. This is a minor new technique we introduced. We then used a Cauchy-Schwartz inequality to bound the covariance to the variance of each classifier. Furthermore, we show the covariance term is upper bounded by the individual error of the classifiers -
> > $\text{Err}\_{\mathcal{D}\_S}(h^*_T)$
> > goes up in the derivation. Intuitively speaking, if the classifier $h^*_T$ has a lower error on the source distribution, then both $h^*_T$ and $h^*_S$ will have a lower error on source $\mathcal{D}_S$ (since $h^*_S$ is optimal on the source distribution), so the two classifiers will overlap more (primarily from the shared examples that they both predicted correctly), and therefore incurs a smaller difference in their induced risks.
> >
> > > Experiments: I don't totally understand the data generation process, even after looking at the supplement. It would be good to have a clearer explanation of the exact strategic modification model here (e.g. what is \epsilon or \alpha, as defined in the supplement)
> >
> > Thanks for the suggestion. We have added more detailed descriptions of the experiments in Appendix C.2.1.
> >
> > > Eq (3 ) it's not clear at this point in the paper if $h^*_T$ is the model with the optimal induced risk, or optimal source risk
> > Good catch; we have added the explanation of $h_T^*$ in equation (3).
> >
> > > Sec 4.2 - should define $X\_+(h)$ and $X\_-(h)$
> >
> > We have added the definition of $X_0+(h)$ and $X_-(h)$ to the main paper.
> >
> > > Sec 5: a little bit confusing to me to overload w here as the induced positive rate, when it was the weighting coefficient earlier
> >
> > Thanks for the suggestion. We have switched $w(h)$ to $p(h)$ to avoid confusion and be consistent with the definition of p.
> >
> > > Thm 5.2: what is p here
> >
> > $p$ represents the ratio of positive instances in the source distribution, namely  $p = Pr_{\mathcal{D}_S}[Y = +1]$.

---

> > ### Comment · Reviewer_tvWE · 2022-11-16
> > **Response**
> >
> > Thanks for the rebuttal. Just to clarify on the question of covariate shift: I'm not sure I quite understand the argument here. In Eq. 3, the assumption states that given X, the probability of some label Y is equal the same value p, regardless of if we are in the source distribution or the shifted distribution. However, suppose we set X to some point just slightly on the positive (h = 1) side of the decision boundary. Then, wouldn't we expect the probability of Y = 1 to be higher in the shifted distribution, since there is a large mass of people who have truly improved their qualifications in order to land on the positive side?

---

> > > ### Author Response · Authors · 2022-11-17
> > > **Followup response to Reviewer tvWE**
> > >
> > > Thank you for your follow-up question. Please allow us to clarify further.
> > >
> > > The example given by the reviewer does not conflict with the condition in our paper:
> > > you are absolutely correct that after the shift, we expect more Y = 1 instances in the shifted distribution because there is a large mass of people who have truly improved their qualifications to land on the positive side. However, this corresponds to having a higher Pr(X = x) and a higher Pr(Y = 1, X = x) near X = x where x is the point that is close to the positive side of the decision boundary, rather than having a higher Pr(Y = 1| X = x).
> > >
> > > To be more specific, according to Bayes' rule, we have Pr(Y = 1, X = x) = Pr(Y = 1| X = x) * Pr(X = x). While both Pr(X =x) (which corresponds to having a large mass of people who have truly improved their qualifications in order to land on the positive side), and Pr(Y = 1, X = x) (which corresponds to observing more Y = 1 near the decision boundary point) can be higher, Pr(Y = 1| X = x), which refers to the underlying mapping between X and Y, remains unchanged, which is oblivious of how much Pr(X) has changed before and after the shift.
> > >
> > > You can think of the assumption Pr(Y|X) is for a fixed individual X. Consider the following example. Let X be the credit score between [200, 800]. Y=1 means being able to repay the loan. What our assumption states is that for anyone whose credit score is 700, the chance of repaying the loan is the same, i.e., P(Y=1|X = 700) is the same. This probability is not the same as the total proportion of people whose credit score is 700, i.e. P(X = 700), nor the proportion of people whose credit score is 700 and has the ability of repaying the loan, i.e. Pr(Y = 1, X = 700).
> > >
> > >
> > > Please let us know if this answers your question, and we would be more than happy to clarify further!

---

> > > > ### Comment · Reviewer_tvWE · 2022-11-17
> > > > **Thanks**
> > > >
> > > > I appreciate the clarification - I understand the distinction you're trying to make now.  I still think that this probably makes some strong assumptions about the decision boundary lining up with the true boundary closely that I'm not sure are realistic, but I can see why this is a useful assumption. In light of this, I'll reconsider my score after discussion with the other reviewers.

---

> > > > > ### Author Response · Authors · 2022-12-03
> > > > > **follow up**
> > > > >
> > > > > Dear Reviewer tvWE,
> > > > >
> > > > > Thank you again for your thoughtful review comments. We'd like to follow up to see if there is any issue that arises during your discussions. We'd love to have an opportunity to address if there is any.
> > > > >
> > > > > Best,
> > > > >
> > > > > Authors

---

> ### Author Response · Authors · 2022-12-09
> **Discussion period will end soon, and your feedback will be appreciated**
>
> Dear Reviewer tvWE,
>
> Hope you are well! Since the discussion period will end soon, we are very much looking forward to your thoughts after your discussion with other reviewers, despite your busy schedule. It will be appreciated if you kindly let us know whether your previous concerns have been properly addressed, and if you have other concerns, we hope for the opportunity to respond to them.
>
> With best wishes, Authors of Paper3030

---

### Official Review · Reviewer_Rs7K · 2022-11-02

**Confidence:** 3
**Correctness:** 3
**Technical Novelty And Significance:** 2
**Empirical Novelty And Significance:** 2
**Recommendation:** 5

**Clarity, Quality, Novelty And Reproducibility:**

Overall the paper is pretty clear. My main concern is still the use of these bounds. In the scenarios described in Section3.3, I would be interested in knowing
1. if the provided bound tight in any sense for the empirical risk minimizing model h_S?
2. as the authors write in Section 6  "indicating the suboptimality of training on D_S", is there any other learning algorithm that provides better bounds? Intuitively, training with a penalty term would likely give better results.

Of course it may involve more assumptions, but I think a tightness result, even a asymmpototic one would be very useful.

**Strength And Weaknesses:**

pros：
the paper is clearly written, and the introduction of the problem is well motivated. The organization of the paper also makes it easy to follow, with clear notations / explanations to the theorems etc.

cons:
As the authors write in Section 3.3, the bounds in the paper is mainly of theoretic interest. I would like to see some disucssions on the tightness of these bounds, even a single example would be helpful. The mathematical tools used are also not deep.


**Summary Of The Paper:**

This paper studies model transferability when human decision subjects respond to the deployed model. It provides a series of lower / upper bound between empirical risk minimizer classifier and the optimal classifier. In particular, teh paper studies two common cases: covariate shift and target shift.

**Summary Of The Review:**

The paper gives a series of bounds for the model transferability, the proofs I checked (non-exhaustive) are all good. It would be nice to see more discussions on the tightness of these bounds, as in the general case, the positiveness of the lower bound is not even clear.

---

> ### Author Response · Authors · 2022-11-13
> **Response to Reviewer Rs7K (1/2)**
>
> Thank you for the feedback. Below we address your comments and questions:
>
> > I would like to see some discussions on the tightness of these bounds, even a single example would be helpful.
>
> This is a great suggestion. Please allow us to provide both empirical and theoretical evidence for the tightness of our bounds.
>
> [Empirical] To further demonstrate the tightness of our bounds, we include additional experiments using synthetic datasets.  In our revision, we conduct additional experiments with synthetic data generated according to the two causal DAGs in our paper (Figure 2) to further validate our bounds (see Appendix C.1 for more details). For all four synthetic datasets, we do observe that two empirical observations we would like to bound (i.e. $\text{Err}({\mathcal{D}(h_S^*)})({h_S^*}) - \text{Err}({\mathcal{D}(h_T^*)})({h_T^*})$
> for upper bound; $\max \{ \text{Err}({\mathcal{D}_S})({h}), \text{Err}({\mathcal{D}(h)})({h}) \}$  for lower bound) are well-bounded by the theoretical results. For the lower bound, the empirical observation and the theoretical bounds are roughly within the same magnitude except for one target shift dataset, indicating the effectiveness of our theoretical result.
>  For the upper bound, under target shift, the empirical observations are well within the same magnitude of the theoretical bounds while the results for the covariate shift are relatively loose.
>
> > In the scenarios described in Section 3.3, I would be interested in knowing if the provided bound tight in any sense for the empirical risk minimizing model h_S?
>
> Continue the discussions above, now we provide some discussions on the tightness of our bounds from the theoretical perspective.
>
> [Theoreticall] For the general bounds reported in Section 3, it is hard to fully quantify the tightness without further quantifying the specific quantities of the terms, e.g. the H-divergence of the source and the induced distribution, and the average error a classifier has to incur for both distributions. This part of our results was adapted from the classical literature on learning from multiple domains [Ben-David et al., 2010a]. The tightness of using $\mathcal H$-divergence and other terms seems to be partially validated therein. A rigorous derivation of the impossibility or tightness results seems to require separate and substantial work [Ben-David et al, 2010b]
> For more specific bounds provided in Section 4 (for covariate shift) and Section 5 (target shift), however, it is relatively easier to argue about the tightness: the proofs there are more straightforward and are easier to back out the conditions where the inequalities are relaxed. For example, in Theorem 5.1, the inequalities of our bound are introduced primarily in the following two places: 1) one is using the optimality of $h^*_S$ on the source distribution. 2) the other is bounding the statistical difference in $h^*_S$ and $h^*_T$’s predictions on the positive and negative examples. Both are saying that if the differences in the two classifiers’ predictions are bounded in a range, then the result in Theorem 5.1 is relatively tight.
> We see this in a controlled synthetic example as reported in Figure 3, Appendix C.2 (we discuss the results in the response above). Overall we observe that the covariate shift lower bounds are tighter. The reason is that the lower bound for target shift captures precisely the changes in the distribution $|p-p(h)|$, as well as the classifier’s baseline accuracy (TPR, FPR) on the distributions that are not shifting.
>
> References:
>
> [Ben-David et.al,  2010a] Shai Ben-David, John Blitzer, Koby Crammer, Alex Kulesza, Fernando Pereira, Jennifer Wortman Vaughan. “A theory of learning from different domains”. Machine Learning 79(1-2): 151-175 (2010)
>
> [Ben-David et.al,  2010b] Ben-David, Shai, Tyler Lu, Teresa Luu, and Dávid Pál. "Impossibility theorems for domain adaptation." In Proceedings of the Thirteenth International Conference on Artificial Intelligence and Statistics, pp. 129-136. JMLR Workshop and Conference Proceedings, 2010.

---

> > ### Author Response · Authors · 2022-11-13
> > **Response to Reviewer Rs7K (2/2)**
> >
> > > “As the authors write in Section 6 "indicating the suboptimality of training on D_S", is there any other learning algorithm that provides better bounds? Intuitively, training with a penalty term would likely give better results”.
> >
> > This is an interesting point. Rederiving the bounds reported in our paper with a proper penalty term doesn’t look super straightforward. The key challenge is to relate the optimal ``penalized” classifier’s expected risk to $h^*_T$, the one that minimizes the induced risk directly. Nonetheless, we empirically tested the idea of penalizing the classifier and observed its difference from $h^*_T$. One possibility is to show that the penalized training leads to an optimizer for a distribution that is closer to the induced target distribution. Then our bounds would imply that the performance gap would be tighter. We have provided such a discussion in Appendix F.
> >
> > In addition, we have carried out experiments to verify the effectiveness of commonly used regularizers. Overall, we do not observe a substantial performance improvement by doing so (details provided in Appendix C.2.2).
> >
> > Notice that the distributional robust algorithm (DRO) would be another promising technique to train a robust classifier that provides better performance on the induced distribution. Here we describe the techniques from DRO. We will construct an uncertainty set around the data-generating distribution and attempt to minimize the expected loss on the worst-case distribution within this uncertainty set. In our setting, the uncertainty set of the source distribution $\mathcal{D}_S$ is defined as $U_f(\mathcal{D}_S) = {\mathcal{D}: D_f(\mathcal{D}||\mathcal{D}_S)\leq \rho}$, where $\mathcal{D}_f$ is a specific statistical distance measurement (e.g., $D_f$ is the KL-divergence), and $\rho$ denotes the upper bound on the statistical difference between the source distribution and any potential induced distributions.
> > The distributionally robust problem is to minimize the prediction error on the worse possible induced distribution. This is equivalent to solving the following minimization problem:
> >
> > $\min\_{\theta\in \Theta} {sup_{\mathcal{D}<< \mathcal{D}_S} E_Q[\ell(\theta; Z)]}$, where $\Theta$ is the parameter space, $\mathcal{D}_S$ is the source distribution, $Z = (X, Y)$ is a random variable generated according to $\mathcal{D}_S$, and $\ell: \Theta \times X \rightarrow \mathcal{R}$ is the loss function.
> > One potential issue using DRO in our setting, however, is that its success of it relies on the construction of the uncertainty set, which corresponds to how accurate the estimation of the statistical difference between $\mathcal{D}_S$ and $\mathcal{D}(h)$ could be. We suspect that having a good approximation of such quantity will help the DRO method provide better bounds overall.
> >
> > References:
> >
> > [Peet-Par et al., 2022] Peet-Pare, L., Hegde, N. and Fyshe, A., 2022. Long Term Fairness for Minority Groups via Performative Distributionally Robust Optimization. arXiv preprint arXiv:2207.05777.
> >
> > [Rahimian et al., 2019] Rahimian, Hamed, and Sanjay Mehrotra. "Distributionally robust optimization: A review." arXiv preprint arXiv:1908.05659 (2019).

---

> > > ### Comment · Reviewer_Rs7K · 2022-11-26
> > > **Thanks for the rebuttal**
> > >
> > > Thanks for the response on these concerns, I have read it , and appreciate the addition of more examples on the tightness as well as the discussion on the regularization part.

---

> > > > ### Author Response · Authors · 2022-12-03
> > > > **thank you**
> > > >
> > > > Dear Reviewer Rs7K,
> > > >
> > > > Thank you. Please kindly let us know there is any last minute issue that we could clarify.
> > > >
> > > > Best,
> > > >
> > > > Authors

---

> ### Author Response · Authors · 2022-12-09
> **Discussion period will end soon, and your feedback will be appreciated**
>
> Dear Reviewer Rs7K,
>
> Hope you are well!  Since the discussion period will end soon, we are very much looking forward to your thoughts after your discussion with other reviewers, despite your busy schedule.  It will be appreciated if you kindly let us know whether your previous concerns have been properly addressed, and if you have other concerns, we hope for the opportunity to respond to them.
>
> With best wishes,
> Authors of Paper3030

---

### Official Review · Reviewer_vuF4 · 2022-11-02

**Confidence:** 3
**Correctness:** 4
**Technical Novelty And Significance:** 3
**Empirical Novelty And Significance:** 2
**Recommendation:** 5

**Clarity, Quality, Novelty And Reproducibility:**

 The related work is, to the best of my knowledge, complete. Regarding discussion, I think a short discussion about the challenge to minimize induced risk should be added in the main paper and it should refer to appendix D for full discussion. The paper is well written. A short paragraph about the difference in setting between domain adaptation and the considered problem could also be useful. In particular, the lack of access to target samples.

**Strength And Weaknesses:**

This paper studies a new problem and defines new quantities to understand and study it. It makes the paper mostly theoretical and related to the statistic field. Authors define many quantities inspired from DA. However as in this problem we do not have access to target samples unlike in DA, this problem seems more related to domain generalization (DG). I wonder if some theoretical or training procedure developed in DG could be applied to this problem and maybe this should be discussed in the paper.

Regarding the bounds, they are explained and applied to several examples. They are also evaluated on the FICO credit score dataset.

**Summary Of The Paper:**

This papers studies the performance of models on the distribution induced by the model itself. The paper first provides examples of when this situation arises. Then, they develop formal definitions of metrics that measures the performance of the model on the induced distributions. These new quantities are inspired by the theory of domain adaptation (DA), like the induced risks that the model should minimize, which is similar to the target risk. Then authors show lower and upper bounds of these quantities. Afterwards, authors use different examples to illustrate their bounds. Finally, their bounds are computed on a real-world dataset.

**Summary Of The Review:**

I agree that the problem is appealing. The defined quantites and theorems are also interesting. The main downside is that I would have expected a training procedure to minimize the induced risk. While authors discuss the challenges of this question in appendix, it should also be discussed in the main paper.

The lack of training procedure to minimize the induced risk and the lack of empirical contributions make that I recommend 'marginally below the acceptance threshold'. I would be happy to increase my score if authors provide additionnal details/toy experiments on how to minimize the induced risk.

---

> ### Author Response · Authors · 2022-11-13
> **Response to Reviewer vuF4 (1/2)**
>
> Thank you for the feedback. Below we address your comments and questions:
>
> >  I wonder if some theoretical or training procedure developed in DG could be applied to this problem and maybe this should be discussed in the paper.
>
> Thank you for pointing out the literature on domain generalization (DG). We provide some discussions comparing our setting with DG in Appendix A.2. In short, the goal of DG is to learn a model that can be generalized to any unseen target distribution. As you point out, one of the biggest challenges in domain generalization is also the lack of target distribution during training. The major difference, however, is that our focus is to understand how the performance of a classifier trained on the source distribution degrades when evaluated on its induced distribution (which depends on how the population of decision subjects responds); In other words, this degradation depends on the classifier itself rather than arbitrary.
> However, we believe that techniques from DG can potentially be applied to our setting when the decision-maker is interested in producing a classifier that will perform well on the induced distribution. In general, we suspect that it will require the decision maker to know certain information about how the induced distribution and the source distribution differ, e.g. some potential characterizations of their statistical differences.  Here we provide two possible directions:
>
> - Data Augmentation: The basic idea of data augmentation is to augment the original $(x, y)$ pairs with new $(A(x), B(y))$ pairs where A and B denote a pair of transformations. Then we add the new pairs to the training dataset.$ A(·)$ and $B(·)$ are usually seen as a way of simulating domain shift, and the design of $A(·)$ and $B(·)$ is key to performance. In our case, A and B are functions of $h$, and they capture how the classifier $h$ influences the potential response from the decision subjects. This requires the decision maker to have a specific response model in mind when designing the augmented data points.
> - Learning Disentangled Representations: Instead of forcing the entire model or features to be domain invariant, which is challenging, one can relax this constraint by allowing some parts to be domain-specific, essentially learning disentangled representations. In our induced domain adaptation setting, this means separating features into two sets, one set contains features that are invariant to the deployment of a classifier $h$, and the other set contains features that will be potentially affected by $h$. Then we can decompose a classifier into domain-specific biases and domain-agnostic weights, and only keep the latter when dealing with the unseen induced domain.
>
> References:
>
> [Zhou et al, 2022] Zhou, K., Liu, Z., Qiao, Y., Xiang, T. and Loy, C.C., 2022. Domain generalization: A survey. IEEE Transactions on Pattern Analysis and Machine Intelligence.
>
> [Sheth et al, 2022] Sheth, Paras, Raha Moraffah, K. Selçuk Candan, Adrienne Raglin, and Huan Liu. "Domain Generalization--A Causal Perspective." arXiv preprint arXiv:2209.15177 (2022).
>
> > I think a short discussion about the challenge to minimize induced risk should be added in the main paper
>
> Thanks for the suggestions. We have added back some discussions about the challenges to the revised main paper.
>
>
> > A short paragraph about the difference in setting between domain adaptation and the considered problem could also be useful. In particular, the lack of access to target samples.
>
> Thank you for the suggestion. We highlight the differences between our setting and other domain adaptation settings (e.g domain generalization, test-time adaptation, adversarial attack) in the related work section, and we discuss the lack of access to target samples during training in the detailed discussion in Appendix A2.

---

> > ### Author Response · Authors · 2022-11-13
> > **Response to Reviewer vuF4 (2/2)**
> >
> > > I would be happy to increase my score if authors provide additional details/toy experiments on how to minimize the induced risk.
> >
> > Thank you for this great suggestion. Please allow us to propose two methods and a toy experiment on how to minimize the induced risk.
> >
> > We first provide a gradient descent based method for a particular label shift setting where the underlying dynamic is the replicator dynamic described in Section 2. Then we propose a solution for a more general case where we do not make any particular assumptions on the underlying distribution shift model. We've added the descriptions of the detailed algorithms, detailed derivations, and additional experimental details in Appendix E.
> >
> > 1. Minimize the induced risk for a label shift replicator dynamic setting:
> >
> > Here we provide a toy example of performing direct induced risk minimization for a particular label shift setting where the underlying agent’s response model is replicator dynamics (as described in section 2.1). The intuition is: under replicator dynamics, we can reformulate the induced risk $E_{\mathcal{D}(h)}(h)$ as a function of only the classifier $h$’s model parameter and other known quantities (e.g. the source distribution $\mathcal{D}_S$, and the utility function, which are fixed), and then perform gradient descent based method on $\theta$ to find an optimal solution, which can be achieved using standard optimization tool (e.g. JAX).
> >
> > Consider a simple setting in which each decision subject is associated with a 1-dimensional continuous feature $x\in \mathcal{R}$ and a binary true qualification $y\in \{-1, +1\}$. Consider a simple threshold classifier $\hat{Y} = h(x) = 1[X \geq \theta]$, meaning that the classifier is completely characterized by the threshold parameter $\theta$. Throughout the process, $P(X|Y = y)$ is sampled according to a known Gaussian distribution and will be unchanged.
> >
> > We first formulate the induced risk as a function of the classifier $h$’s parameter $\theta$ taking into account the underlying dynamic and then perform gradient descent to solve for locally optimal classifier $h_T^*$.
> >
> > Recall from Section 5, under label shift, we can write the induced risk as:
> >
> > $\mathbb{E}\_{D(h)}[\ell(h;X,Y)] = P_{\mathcal{D}(h)}(Y = +1)\cdot \mathbb{E}_{\mathcal{D}_S}[\ell(h;X,Y)|Y=+1] + (1- P\_{D(h)}(Y = +1)) \cdot \mathbb{E}\_{D_S}[\ell(h;X,Y)|Y=-1]$
> >
> > Notice that $\mathbb{E}\_{\mathcal{D}\_S}[\ell(h;X,Y)|Y=+1]$ and $\mathbb{E}\_{\mathcal{D}_S}[\ell(h;X,Y)|Y=-1]$ are already functions of only $h$ and $\mathcal{D}_S$, it suffices to show that the accuracy on $\mathcal{D}(h)$, $P\_{\mathcal{D}(h)}(Y = +1)$, can also be expressed as a function of $\theta$ and $\mathcal{D}_S$. This can be shown by expressing the accuracy on $\mathcal{D}(h)$ as the sum of the prediction accuracy for each $Y = y$, which can further be written as a function of the fitness for each $Y = y$, which is a function of only the source distribution, as well as the utility function, both of them, are pre-specified before the adaptation. Please refer to Section E.1. for more detailed derivations. Then we can use perform gradient descent on the induced risk using automatic differentiation w.r.t $\theta$ and find the locally optimal solution.
> >
> > For the experimental result, we observe that for each setting, the proposed gradient-based optimization procedure returns a classifier that achieves better-induced risk compared to the induced risk of $h^*_S$, which corresponds to minimizing the source distribution.
> >
> > 2. General optimization procedure for (convex) induced risk minimization:
> >
> > In general, finding the optimal classifier that achieves the optimal induced risk $h^*_T$ is a hard problem due to the interactive nature of the problem (see, e.g. the literature on performative prediction [Perdomo et al. 2020]  for a more detailed discussion). Without making any assumptions on the mapping between $h$ and $\mathcal{D}(h)$, one can only rely on the bandit feedback from the decision subjects to estimate the influence of $h$ on its induced data distribution $\mathcal{D}(h)$. Specifically, when the induced risk is a convex function of the classifier h’s parameter $\theta$, we can use the standard techniques from bandit optimization [Flexman et al. 2004] to iteratively find induced optimal classifier $h^*_T$.
> >
> > The basic idea is: at each step $t = 1, \cdots, T$, the decision maker deploys a classifier $h_t$ (characterized by the model parameter $\theta_t$), then observe data points $z_1, \cdots z_{n_t}\sim \mathcal{D}(h_t)$ and their corresponding losses $\ell(z_i; \theta_t)$, and use them to construct an approximate gradient for the induced risk as a function of the model parameter $\theta_t$ Then we perform gradient descent based on this estimated gradient.
> >
> > We describe the detailed algorithm in Appendix E.2.  When the induced risk is a convex function in the model parameter $\theta$, one can show that the above approach guarantees to converge to $h_T^*$.

---

> > > ### Comment · Reviewer_vuF4 · 2022-11-17
> > > **Thank you for your comments.**
> > >
> > > I have read the rebuttal. I appreciate the different toy examples as well as the added literature on domain generalization. I have no further questions and I will reconsider my score after the discussion with the other reviewers.

---

> > > > ### Author Response · Authors · 2022-12-03
> > > > **follow up**
> > > >
> > > > Dear Reviewer vuF4,
> > > >
> > > > Thank you again for your thoughtful review comments. We'd like to follow up to see if there is any issue that arises during your discussions. We'd love to have an opportunity to address if there is any.
> > > >
> > > > Best,
> > > >
> > > > Authors

---

> ### Author Response · Authors · 2022-12-09
> **Discussion period will end soon, and your feedback will be appreciated**
>
> Dear Reviewer vuF4,
>
> Hope you are well!  Since the discussion period will end soon, we are very much looking forward to your thoughts after your discussion with other reviewers, despite your busy schedule.  It will be appreciated if you kindly let us know whether your previous concerns have been properly addressed, and if you have other concerns, we for the opportunity to respond to them.
>
> With best wishes,
> Authors of Paper3030

---

### Official Review · Reviewer_Cedq · 2022-11-04

**Confidence:** 3
**Correctness:** 3
**Technical Novelty And Significance:** 4
**Empirical Novelty And Significance:** Not applicable
**Recommendation:** 8

**Clarity, Quality, Novelty And Reproducibility:**


The writing is good. No apparaent typos are noticed. The authors supplies sufficient materials (codes) for reproducing the experimental results in this paper.


**Strength And Weaknesses:**


### Strength

1. The setting studied in this paper is important, interesting, and novel.
2. This paper studies most fundamental questions in this setting and presents satisfactory results that covers several fundamental lower and upper bounds in this setting.
3. As a mainly theoretical paper, this paper is aware of the practical impact of their theoretical results. To this end, this paper shows how to compute their upper bounds in real-world problems and conducts experiments to demonstrate their results. Furthermore, the experimental results corroborates their theoretical results and suggests that the theoretical results could give meaningful upper and lower bounds for the transfer risks.

### Weakness

1. The proof techniques look simple and the techniques themselves might not inspire broader community.
2. The authors could possibly conduct experiments on more datasets to strengthen their experimental results.


**Summary Of The Paper:**


This paper formulates a very interesting and novel problem (IDA, induced domain adapatation) in transfer learning. Consider the supervised classification setting where one usually trains a classifier $h : X \mapsto Y$ from samples $\{(X, Y)\}$ drawn from some distribution $\sim \mathcal D$. Oftentimes, when the data are generated from human input, the human could possibly modify $(X, Y)$ to adapt to $h$, resulting in a distribution shift over $\mathcal D$. Therefore, when training $h$, it is important to take this distribution shift into consideration. However, this can become complicated and interactive if the human further adapt to $h$.

This paper conducts a rather detailed study on this problem. It proves upper and lower bounds for the transfer risks for several important fundamental questions in this setting, as outlined in page 2. Besides, the paper realizes their bounds by both showing how to compute them in practice and computing them on real datasets.


**Summary Of The Review:**

This paper has good theoretical and empirical results, yet the techniques might not be exciting and the experiments might not be convincing enough due to paucity of datasets evaluated.

---

> ### Author Response · Authors · 2022-11-13
> **Response to Reviewer Cedq**
>
> Thank you for the feedback. Below we address your comments and questions:
>
> > The authors could possibly conduct experiments on more datasets to strengthen their experimental results.
>
> We totally agree with you that there could be more extensive empirical studies done in our work.  In our revision, we conduct additional experiments with synthetic data generated according to the two causal DAGs in our paper (Figure 2) to further validate our bounds (see Appendix C.1 for more details). For all four synthetic datasets, we do observe that two empirical observations we would like to bound (i.e. $\text{Err}({\mathcal{D}(h_S^*)})({h_S^*}) - \text{Err}({\mathcal{D}(h_T^*)})({h_T^*})$
> for upper bound; $\max \{ \text{Err}({\mathcal{D}_S})({h}), \text{Err}({\mathcal{D}(h)})({h}) \}$  for lower bound) are well-bounded by the theoretical results. For the lower bound, the empirical observation and the theoretical bounds are roughly within the same magnitude except for one target shift dataset, indicating the effectiveness of our theoretical result. For the upper bound under target shift, the empirical observations are also well within the same magnitude of the theoretical bounds while the results for the covariate shift are relatively loose. In addition, we also provide new results that perform direct induced risk minimization (as suggested by reviewer vuF4, in Appendix E), as well as training using penalty terms (as suggested by reviewer Rs7K, in Appendix C.2.2).
>
> In general, as far as we know, there is no benchmark dataset clearly suitable for the induced distribution shift study at this point. Most of the standard datasets in domain adaptation (e.g. Office-31, VisDA) might not be suitable for our case, since these adaptations aren’t necessarily the result of the deployment of any particular classifier; thus performing experiments on those standard datasets might not necessarily provide useful empirical support and verification of our theorems. If you are aware of any more related datasets, please let us know, and we are more than happy to include more empirical studies in our paper.

---

> > ### Comment · Reviewer_Cedq · 2022-11-24
> > **Acknowledgement of Rebuttal**
> >
> > Thank the authors for their rebuttal.

---

### Author Response · Authors · 2022-11-13
**Summary of the rebuttal**



We thank all of our reviewers for all the detailed and constructive feedback! Please allow us to provide a summary of the main rebuttal here, followed by a summary of our revisions. We also respond to our reviewers individually with separate, official comments.

**Summary of Rebuttal**

Here we provide a summary of the main contents of this rebuttal:
- Including more synthetic experiments to further validate our bounds (Reviewer Cedq, Reviewer Rs7K)
- Discussions and proposed methods on how to minimize the induced risk (Reviewer vuF4)
- Relate our work to the literature of domain generalization (Reviewer vuF4)
- Discussion of the tightness of our bounds (Reviewer Rs7K)
- Explanation of multiple assumptions made in the papers (Reviewer tvWE)


**Summary of the revisions**

Here we provide a summary of a list of changes we made in the revised paper. Due to the page limit, most of the newly added discussions are added to the appendix.
- Add additional synthetic control experiments in Appendix D2 (Reviewer Cedq, Reviewer Rs7K)
- Add a short discussion about the challenge to minimize induced risk in the main paper (Reviewer vuF4)
- Add more detailed explanations on replicator dynamics in Section 2.
- Add discussions on the relationships of our induced domain adaptation and other domain adaptation settings in Section 1.1 and Appendix A2 (Reviewer vuF4)
- Clarify the assumption made in the strategic classification as an example of covariate shift in Section 4.2 (Reviewer tvWE)
- Add additional descriptions of the experimental setup in the main content. (Reviewer tvWE)
- Add discussion on how to directly induced risk minimization and the corresponding toy experiments in Appendix E (Reviewer Rs7K, Reviewer vuF4)
- Add discussion on adding penalty terms for our training objective in Appendix F (Reviewer Rs7K)
- Add discussions of the tightness of our bounds in Appendix G (Reviewer Rs7K)
Fix confusing notations, typos, and other minor issues (from all reviewers)

---

### Decision · Program_Chairs · 2023-01-20

**Decision:**

Reject

**Justification For Why Not Higher Score:**

The contributions are interesting but too incremental

**Justification For Why Not Lower Score:**

N/A

**Metareview: Summary, Strengths And Weaknesses:**

This paper looks at the following property. When a ML algorithm outputs something (a classifier or whatever), the environment reacts to it and this induces a change in the data-generating process. As a consequence, the expectation of the loss after the output is not the same before. This paper basically looks at how this change occurs and quantifies it.


This idea of response to output is not new and is precisely the purpose and motivation of performative predictions. I find it weird that this paper only discuss in a few lines the differences with it. I would have expected a lot more. So I cannot give credit to that paper of this idea.

Then, the reviewers are also lukewarm concerning the other contributions (the proofs are certainly not straightforward, but techniques are quite common) and I have to agree with them.

In conclusion, this is a nice paper, but too incremental to reach the high bar of ICLR. This said, I am pretty confident that it will face a better fate once revised and resubmitted elsewhere.